# Unraveling the role of Ctla-4 in intestinal immune homeostasis through a novel Zebrafish model of inflammatory bowel disease

Lulu Qin[1], Chongbin Hu[1], Qiong Zhao[1], Yong Wang[1], Dongdong Fan[1], Aifu Lin[1], Lixin Xiang[1]*, Ye Chen[1,2]*, Jianzhong Shao[1,3]*

[1]College of Life Sciences, Key Laboratory for Cell and Gene Engineering of Zhejiang Province, Zhejiang University, Hangzhou, China; [2]Department of Genetic and Metabolic Disease, the Children's Hospital, Zhejiang University School of Medicine, National Clinical Research Center for Child Health, Hangzhou, China; [3]Laboratory for Marine Biology and Biotechnology, Qingdao National Laboratory for Marine Science and Technology, Qingdao, China

*For correspondence:
xianglx@zju.edu.cn (LX);
yechency@zju.edu.cn (YC);
shaojz@zju.edu.cn (JS)

Competing interest: The authors declare that no competing interests exist.

## eLife Assessment

This study focuses on the role of a T-cell-specific receptor, ctla-4, in a new zebrafish model of IBD-like phenotype. Although implicated in IBD diseases, the function of ctla-4 has been hard to study in mice as the KO is lethal. Ctla-4 mutant zebrafish exhibited significant intestinal inflammation and dysbiosis, mirroring the pathology of inflammatory bowel disease (IBD) in mammals, providing a new **valuable** model to the field of IBD research. This is a key study with **convincing** evidence, comprehensive transcriptomic analysis, histological examinations, and functional assays all supporting the findings.

**Abstract** Inflammatory bowel disease (IBD) is a chronic and relapsing immune-mediated disorder characterized by intestinal inflammation and epithelial injury. The underlying causes of IBD are not fully understood, but genetic factors have been implicated in genome-wide association studies, including CTLA-4, an essential negative regulator of T cell activation. However, establishing a direct link between CTLA-4 and IBD has been challenging due to the early lethality of CTLA-4 knockout mice. In this study, we identified zebrafish Ctla-4 homolog and investigated its role in maintaining intestinal immune homeostasis by generating a Ctla-4-deficient (ctla-4$^{-/-}$) zebrafish line. These mutant zebrafish exhibited reduced weight, along with impaired epithelial barrier integrity and lymphocytic infiltration in their intestines. Transcriptomics analysis revealed upregulation of inflammation-related genes, disturbing immune system homeostasis. Moreover, single-cell RNA-sequencing analysis indicated increased Th2 cells and interleukin 13 expression, along with decreased innate lymphoid cells and upregulated proinflammatory cytokines. Additionally, Ctla-4-deficient zebrafish exhibited reduced diversity and an altered composition of the intestinal microbiota. All these phenotypes closely resemble those found in mammalian IBD. Lastly, supplementation with Ctla-4-Ig successfully alleviated intestinal inflammation in these mutants. Altogether, our findings demonstrate the pivotal role of Ctla-4 in maintaining intestinal homeostasis. Additionally, they offer substantial evidence linking CTLA-4 to IBD and establish a novel zebrafish model for investigating both the pathogenesis and potential treatments.

## Introduction

IBD, including Crohn's disease and ulcerative colitis, refers to a group of chronic relapsing inflammation disorders affecting the gastrointestinal tract, that have been increasing in prevalence worldwide (*Hodson, 2016*). The precise etiology of IBD has yet to be fully elucidated. Conventional epidemiological studies have indicated that IBD tends to run in families and is linked to genetic factors (*Zhang and Li, 2014*; *Uniken Venema et al., 2017*). However, research also suggested that susceptibility gene patterns differ significantly among various geographic populations. Current evidence points towards a complicated interaction involving host genetics, disrupted intestinal microbiota, environmental triggers, and abnormal immune responses (*Loftus, 2004*; *Khor et al., 2011*; *Neurath, 2020*). Advances in genomic sequencing techniques have allowed for the identification of genetic variants associated with an increased risk of developing IBD. Among these, mutations in immune-related genes have received particular attention. Research on humans with Crohn's disease and mouse models of IBD has shown that genetically susceptible individuals exhibit defects in intracellular pattern-recognition receptors (PRRs), such as toll-like receptors (TLR) and nucleotide-binding oligomerization domain (NOD)-like receptors (NLRs), which are responsible for initiating innate immune responses to eliminate harmful bacteria (*Kordjazy et al., 2018*; *Horowitz et al., 2021*). Genetic variations in the tumor necrosis factor ligand superfamily member 15 (TNFSF15) and interleukin 23 receptor (IL23R) genes, both involved in suppressing inflammation, have been associated with an increased risk of developing Crohn's disease (*Duerr et al., 2006*; *Tremelling et al., 2008*).

Cytotoxic T lymphocyte antigen-4 (CTLA-4), also known as CD152, is one of the most well-established immune checkpoint molecules expressed predominantly on T cells (*Rudd and Schneider, 2003*; *Yang et al., 2021*; *Kim and Choi, 2022*). It primarily regulates the early stages of T-cell activation by attenuating downstream signaling of the T-cell receptor (TCR) (*Marengère et al., 1996*; *Lee et al., 1998*; *Yokosuka et al., 2010*). Specifically, CTLA-4 has a much higher affinity for CD80 and CD86 ligands compared to the co-stimulatory receptor CD28 (*Linsley et al., 1994*; *Collins et al., 2002*). By outcompeting CD28 for ligand binding, CTLA-4 provides an inhibitory signal that impacts immunological synapse formation and inhibits T-cell proliferation and activation (*Yokosuka et al., 2010*; *Saito et al., 2010*). The immunomodulatory role of CTLA-4 in maintaining immune homeostasis is highlighted through CTLA-4 knockout studies. Germline CTLA-4-deficient is lethal for mice within 3–4 wk due to massive T lymphocyte proliferation and the release of inflammatory cytokines (*Tivol et al., 1995*; *Waterhouse et al., 1995*; *Chambers et al., 1997*). Compared to wild-type T lymphocytes, CTLA-4-deficient T lymphocytes exhibit accelerated development of Th2 cells, leading to significantly enhanced secretion of IL-4 and IL-5 (*Khattri et al., 1999*; *Bour-Jordan et al., 2003*). Additionally, conditional deletion of CTLA-4 in adult mice results in rapid immune activation, multiorgan lymphocyte infiltration, and autoantibody production (*Klocke et al., 2016*). Moreover, a selective deficiency of CTLA-4 in Treg cells is sufficient to induce lymphoproliferation and autoimmune diseases in mice (*Wing et al., 2008*). Similarly, associations between polymorphic alleles of CTLA-4 and IBD in humans have been reported in multiple studies (*Repnik and Potocnik, 2010*; *Xia et al., 2002*; *Jiang et al., 2010*). Moreover, CTLA-4 is an intriguing target for novel immune checkpoint blockade therapies in cancer treatment, while intestinal inflammation is a common side effect in these clinical trials (*Bamias et al., 2017*; *Lo et al., 2024*). Establishing a direct causal relationship between CTLA-4 and IBD has been challenging due to difficulties in finding appropriate models. The early lethality observed in CTLA-4-deficient mice added another layer of complexity to this issue. Zebrafish is a powerful model system for immunological and biomedical research, due to its versatility and high degree of conservation in innate and adaptive immunities (*Cooper and Alder, 2006*; *Lam et al., 2004*). In our current study, we identified the Ctla-4 homology in zebrafish and successfully developed an adult vertebrate model with homozygous knockout of the *ctla-4* gene for the first time. These *ctla-4*-deficient (*ctla-4⁻/⁻*) zebrafish survive but exhibit attenuated growth and weight loss. Notably, *ctla-4* deficiency leads to an IBD-like phenotype in zebrafish characterized by altered intestinal epithelial cell morphology, abnormal inflammatory response, defects in microbial stratification, and composition. Mechanistically, Ctla-4 exerts its inhibitory function by competing with Cd28 for binding to Cd80/86. These findings establish the *ctla-4* knockout zebrafish as an innovative platform to elucidate CTLA-4 immunobiology, model human IBD, and develop novel therapeutic modalities.

## Results

### Identification of zebrafish Ctla-4

Through a homology search in the NCBI database, we identified the *ctla-4* gene (XM_005167519.4) on zebrafish chromosome 9, which exhibits an exon organization comparable to that of the human CTLA-4 gene (*Figure 1—figure supplement 1A–C*). Zebrafish Ctla-4 is predicted to be a type I transmembrane protein with a molecular weight of approximately 33 kDa, featuring structural characteristics of the immunoglobulin superfamily, which include an N-terminal signal peptide, a single IgV-like extracellular domain, a transmembrane region, and a cytoplasmic tail (*Figure 1A*). Multiple amino acid sequence alignments revealed that Ctla-4 contains a [113]LFPPPY[118] motif within the ectodomain and a tyrosine-based [206]YVKF[209] motif in the distal C-terminal region (*Figure 1A*). These motifs closely resemble the MYPPPY and YVKM motifs found in mammalian CTLA-4 homologs, which are essential for binding to CD80/CD86 ligands, as well as molecular internalization and signaling inhibition (*Marengère et al., 1996*; *Peach et al., 1994*; *Shiratori et al., 1997*). The IgV-like domain of Ctla-4 was characterized by a two-layer β-sandwich and was conserved between zebrafish and humans (*Figure 1B*). In contrast, zebrafish Cd28 features a SYPPPF motif in its extracellular region and a FYIQ motif in its intracellular tail, distinguishing it from Ctla-4. Additionally, zebrafish Ctla-4, similar to its counterparts in other species, carries a conserved extracellular [123]GNGT[126] motif, which is absent in zebrafish Cd28 (*Bernard et al., 2006*). This structural distinction further differentiates Ctla-4 from Cd28 (*Figure 1A*, *Figure 1—figure supplement 1D*). Consistent with this, phylogenetic analysis showed that Ctla-4 clusters with other known CTLA-4 homologs from different species with high bootstrap probability, whereas zebrafish Cd28 groups separately with other CD28s (*Figure 1—figure supplement 1C*). Structurally, Ctla-4 exists as a dimer, and unlike the intracellular localization of CTLA-4 in mammals, Ctla-4 is found on the cell membrane (*Figure 1C and D*). By analyzing the splenic scRNA-seq dataset we recently established (*Hu et al., 2023*), Ctla-4 was primarily expressed on the T cells, including the Cd4[+] T and Cd8[+] T cells (*Figure 1E*). This result was verified by immunofluorescence assays on the splenic leukocytes (*Figure 1F*).

### Ctla-4 deficiency induces inflammatory bowel disease (IBD)-like phenotype

To further investigate the function of Ctla-4, we generated a *ctla-4[-/-]* zebrafish line with a 14-base deletion in the second exon of the *ctla-4* gene (*Figure 2A–C*). The zebrafish appeared grossly normal in appearance; however, the body weight and size were significantly reduced compared with those of wild-type zebrafish (*Figure 2D and E*). Anatomically, the *ctla-4[-/-]* zebrafish were featured by intestine shortening and splenomegaly, suggesting the occurrence of chronic inflammation in the intestines (*Figure 2F and G*). For clarification, we first performed histological analysis on the anterior, mid, and posterior intestine segments using H&E staining. Compared to the wild-type zebrafish, the *ctla-4[-/-]* fish exhibited significant epithelial hyperplasia in the anterior intestine segment, accompanied with a small amount of mucosal inflammatory cell infiltration (*Figure 2H*). Moreover, noteworthy goblet cell loss, reduction of normal mucins, and the accumulation of acidic mucins were also observed in *ctla-4[-/-]* anterior intestine, as detected through Alcian Blue and Periodic Acid-Schiff (AB-PAS) or PAS staining (*Figure 2I and J*, *Figure 2—figure supplement 1A and B*). A small amount of lymphocytic infiltration and mild epithelial damage occurred in the mid-intestine segment of *ctla-4[-/-]* zebrafish (*Figure 2H*). In the posterior intestine of *ctla-4[-/-]* fish, the intestinal villi were markedly shortened, the epithelial barrier showed severely disrupted, and the intestinal wall became thinner, wherein the mucosal and transmural inflammatory cells were significantly infiltrated (*Figure 2H*). Notably, the intestinal lumens in all three intestinal segments were enlarged in the ctla-4[-/-] zebrafish, and the ratio between the length of the intestinal villi and the intestinal ring radius was higher in the ctla-4[-/-] zebrafish intestines compared to those in wild-type zebrafish (*Figure 2—figure supplement 1C*). In addition, the ultrastructure analysis revealed that the epithelial cells of posterior intestine in *ctla-4[-/-]* zebrafish exhibited alteration in tight junction, the loss of adhesion junctions and desmosomes, and disruption of microvilli (*Figure 2K*). All these results strongly indicate that Ctla-4 plays a crucial role in preserving intestinal homeostasis in zebrafish. The intestinal phenotype resulting from Ctla-4 deficiency was similar to IBD in mammals.

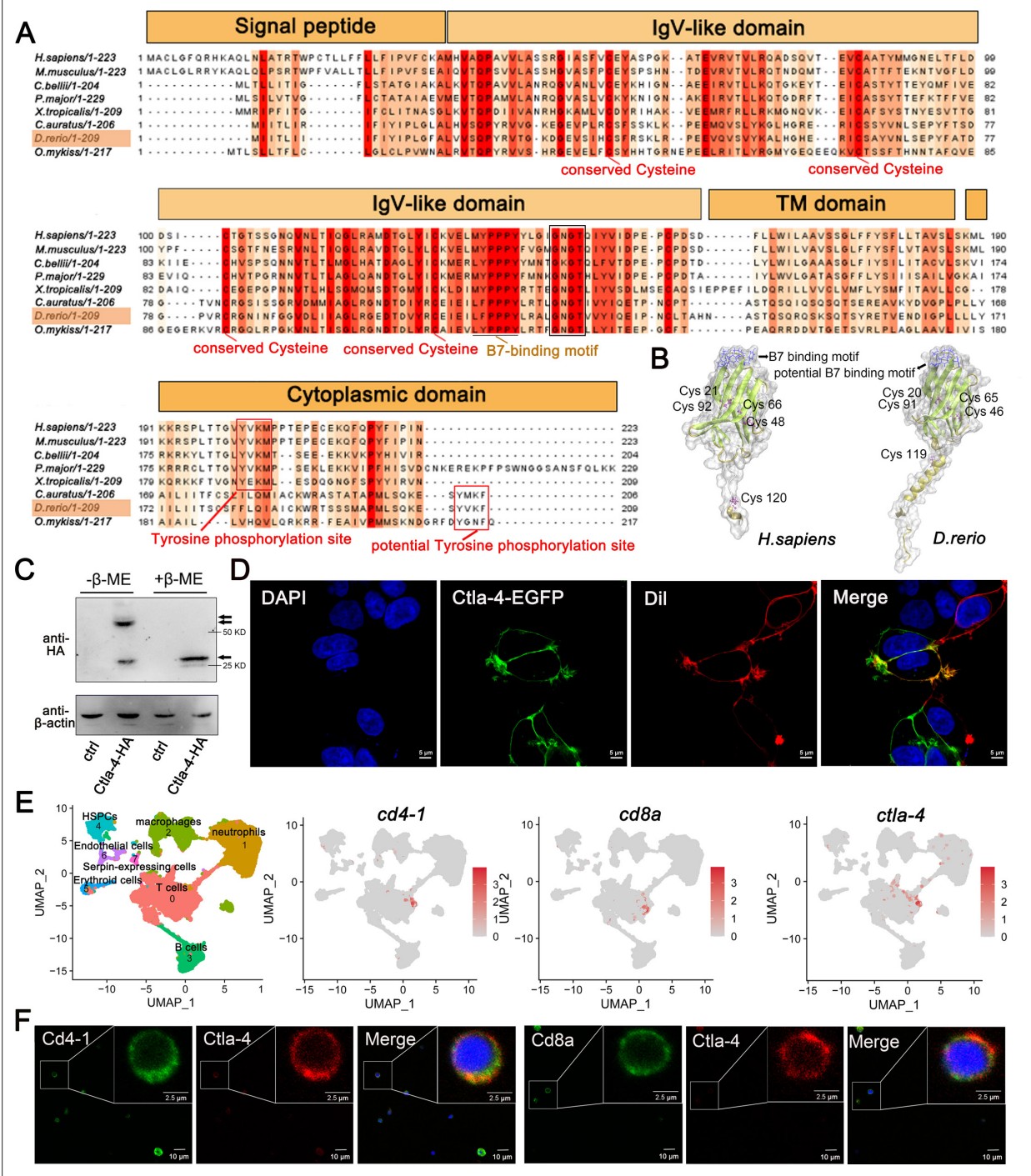

**Figure 1.** Characterization of zebrafish Cytotoxic T lymphocyte antigen-4 (Ctla-4). (**A**) Alignment of the Ctla-4 homologs from different species generated with ClustalX and Jalview. The conserved and partially conserved amino acid residues in each species are colored in hues graded from orange to red, respectively. Key features, including conserved cysteine residues, functional motifs, such as B7-binding motif, tyrosine phosphorylation site, and potential tyrosine phosphorylation site, were indicated separately. The signal peptide, IgV-like domain, transmembrane (TM) domain, and cytoplasmic domain were marked above the sequence. (**B**) The tertiary structure of the zebrafish Ctla-4 ectodomain, as predicted by AlphaFold2, was compared with that of humans. The two pairs of disulfide bonds (Cys$^{20}$-Cys$^{91}$/Cys$^{46}$-Cys$^{65}$ in zebrafish and Cys$^{21}$-Cys$^{92}$/Cys$^{48}$-Cys$^{66}$ in humans) used to connect the two-layer β-sandwich, and the separate Cys residue (Cys$^{119}$ in zebrafish and Cys$^{120}$ in humans) involved in the dimerization of the proteins are indicated. Cysteine residues are represented in purple ball-and-stick models, and the identified or potential B7 binding sites are highlighted in blue. (**C**) Dimer of Ctla-4 was identified by Western blot under reducing (+β-ME) or non-reducing (-β-ME) conditions. The ctrl represents a control sample derived from cells transfected with an empty plasmid. The monomers and dimers were indicated by single and double arrows, respectively. (**D**) The subcellular

*Figure 1 continued on next page*

Figure 1 continued

localization of Ctla-4 protein was assessed in HEK293T cells transfected with pEGFPN1-Ctla-4 for 48 hr, imaged using a two-photon laser-scanning microscope (Original magnification, 630×). Nuclei were stained with DAPI (blue), and cell membranes were stained with DiI (red). (**E**) UMAP plots showing the relative distribution of common T cell markers (*cd4-1*, *cd8a*, and *ctla-4*) based on a splenic single-cell RNA sequencing (scRNA-seq) dataset we recently established (*Hu et al., 2023*). (**F**) Immunofluorescence staining of lymphocytes isolated from zebrafish blood, spleen, and kidney. Cells were stained with mouse anti-Ctla-4, together with rabbit anti-Cd4-1 or rabbit anti-Cd8α. DAPI stain shows the location of the nuclei. Images were obtained using a two-photon laser-scanning microscope (Original magnification, 630×).

The online version of this article includes the following source data and figure supplement(s) for figure 1:

**Source data 1.** Source data for *Figure 1C*.

**Source data 2.** Source data for *Figure 1C*.

**Figure supplement 1.** The organization, sequence, and phylogenetic analysis of zebrafish Cytotoxic T lymphocyte antigen-4 (*ctla-4*) and *cd28* genes.

**Figure supplement 2.** Preparation of mouse anti-Ctla-4 antibody.

**Figure supplement 2—source data 1.** Source data for *Figure 1—figure supplement 2A–C*.

**Figure supplement 2—source data 2.** Source data for *Figure 1—figure supplement 2A–C*.

## Molecular mechanisms of Ctla-4 deficiency-induced IBD-like phenotype

To explore the potential molecular mechanisms of Ctla-4 deficiency-induced IBD-like phenotype, we performed transcriptome profiling analysis of intestines from wild-type and *ctla-4*$^{-/-}$ zebrafish. We identified a total of 1140 differentially expressed genes (DEGs), among which 714 genes were up-regulated, and 426 genes were down-regulated in *ctla-4*$^{-/-}$ zebrafish (*Figure 3A and B*). GO enrichment analysis showed that DEGs or up-regulated genes in the top 10 enriched biological processes were associated with the immune response and inflammatory response (*Figure 3C and D*). Moreover, the KEGG enrichment analyses indicated that the up-regulated DEGs are primarily involved in the process of cytokine-cytokine receptor interaction and cell adhesion molecules, which are also related to inflammation (*Figure 3—figure supplement 1A*); however, the down-regulated DEGs were significantly enriched in the process of metabolism (*Figure 3—figure supplement 1B*). The intestines of *ctla-4*$^{-/-}$ zebrafish showed significant upregulation of *il1b*, *tnfa*, myeloid-specific peroxidase (*mpx*), matrix metallopeptidase 9 (*mmp9*), chemokine (C-X-C motif) ligand 8 a (*cxcl8a*), and *il13*. In contrast, *il10*, a potent anti-inflammatory cytokine, was markedly down-regulated in Ctla-4-deficient intestines (*Figure 3E*). The transcriptional change of these genes was confirmed by RT-qPCR (*Figure 3F*). By constructing the protein-protein interaction (PPI) network, we found that *il1b* was a major cytokine that played a hub role in promoting the bowel inflammation of *ctla-4*$^{-/-}$ zebrafish (*Figure 3G*). Moreover, Gene set enrichment analysis (GSEA) showed that genes involved in the lymphocyte chemotaxis, positive regulation of ERK1 and ERK2 cascade, Calcium and MAPK signaling pathways were positively enriched in *ctla-4*$^{-/-}$ zebrafish intestines, implying a sensitized or activated state of lymphocytes due to the absence of Ctla-4 (*Figure 3—figure supplement 1C and D*). Notably, biological processes related to neutrophil activation and chemotaxis were significantly enriched (*Figure 3C and D*). Studies have shown that neutrophils can induce histopathological effects through releasing matrix metalloproteinases (MMPs), neutrophil elastase, and myeloperoxidase (MPO) (*Butin-Israeli et al., 2019*). To confirm the association between neutrophils and Ctla-4-deficient intestinal inflammation, the MPO level was examined. As a support, MPO activity was markedly increased in the intestines and peripheral blood of *ctla-4*$^{-/-}$ zebrafish (*Figure 3H*). Besides, a number of biological markers or susceptibility genes of IBD observed in mammals, including c-reactive protein 6 (*crp6*), *crp7*, MMPs, haptoglobin, *il23r*, insulin-like growth factor binding protein 1 a (*igfbp1a*), cAMP responsive element modulator b (*cremb*), and lymphocyte specific protein 1 b (*lsp1b*), were highly expressed in the *ctla-4*$^{-/-}$ zebrafish (*Figure 3I and J*; *Duerr et al., 2006*; *Lees et al., 2011*; *Lee et al., 2017*), suggesting the presence of a conserved molecular network underlying IBD pathogenesis between *ctla-4*$^{-/-}$ zebrafish and mammalian models.

## Cellular mechanisms of Ctla-4 deficiency-induced IBD-like phenotype

To investigate the cellular mechanisms underlying the IBD-like phenotype induced by Ctla-4 deficiency, we performed scRNA-seq analysis on intestines from wild-type and *ctla-4*$^{-/-}$ zebrafish using the 10×Genomics Chromium platform. We obtained nine discrete clusters from 7,539 cells of wild-type and *ctla-4*$^{-/-}$ zebrafish (*Figure 4A*). These clusters of cells were classified as enterocytes, enteroendocrine cells, smooth muscle cells, neutrophils, macrophages, B cells, and a group of T/NK/ILC-like

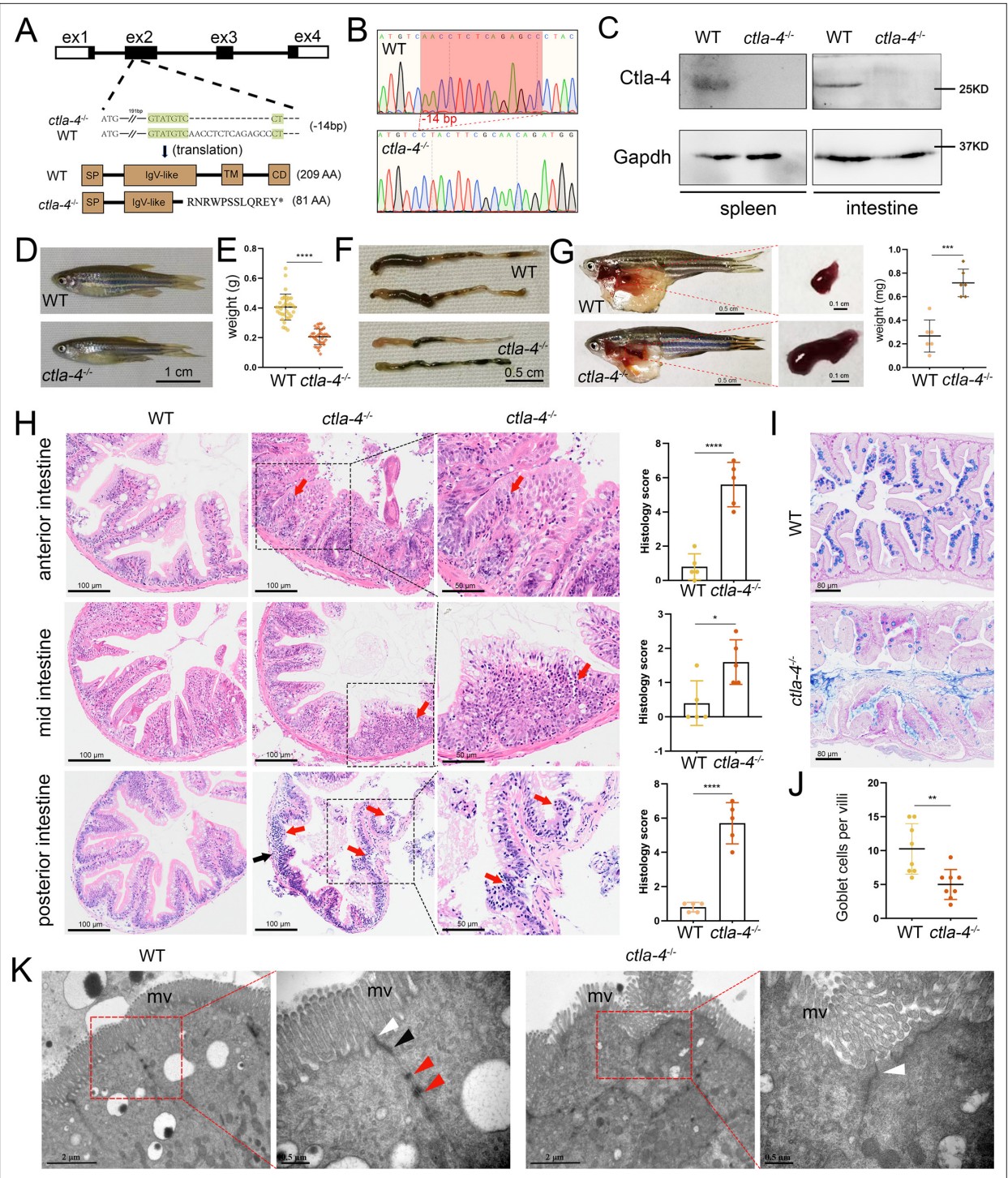

**Figure 2.** Examination on the inflammatory bowel disease (IBD)-like phenotype in *ctla-4*⁻/⁻ zebrafish. (**A**) Generation of a homozygous Cytotoxic T lymphocyte antigen-4 (*ctla-4*)-deficient (*ctla-4*⁻/⁻) zebrafish line through CRISPR/Cas9-based knockout of *ctla-4* gene on chromosome 9. A 14 bp deletion mutation in exon 2 results in a premature stop at codon 82, which is predicted to produce a defective Ctla-4 protein containing 81 amino acids. (**B**) Genotyping of the deficiency of *ctla-4* gene by Sanger sequencing. (**C**) Knockout efficiency of Ctla-4 selectively examined in spleen and gut tissues of *ctla-4*⁻/⁻ zebrafish by Western blot analysis. Gapdh serves as a loading control. (**D**) Normal gross appearance of adult wild-type (WT) and *ctla-4*⁻/⁻ zebrafish. (**E**) Body weight statistics of WT and *ctla-4*⁻/⁻ zebrafish (n=30). (**F**) The change of intestine length in WT and *ctla-4*⁻/⁻ zebrafish. (**G**) The change of splenic size in WT and *ctla-4*⁻/⁻ zebrafish. (**H**) Representative H&E staining analysis of histopathological changes and quantitation of histology scores in the anterior, mid, and posterior intestines from WT and *ctla-4*⁻/⁻ zebrafish. Red arrows denote mucosal inflammatory cell infiltration, and black arrow indicates transmural inflammatory cell infiltration. (**I**) Alcian Blue and Periodic Acid-Schiff (AB-PAS) staining was used to analyze the mucin components

*Figure 2 continued on next page*

*Figure 2 continued*

and the number of goblet cells in anterior intestine from WT and *ctla-4*[-/-] zebrafish (n=5). (J) Quantitation analysis of goblet cells of each villus in the foregut of WT and *ctla-4*[-/-] zebrafish (n=8). (K) Observation of cell junctions between intestinal epithelial cells in posterior intestines from WT and *ctla-4*[-/-] zebrafish under TEM (Hitachi Model H-7650). White triangles indicate tight junctions, black triangles indicate adhesion junctions, and red triangles indicate desmosomes. Data are presented as mean ± standard deviation (SD). Statistical significance was assessed through an unpaired Student's t-test (*$p < 0.05$; **$p < 0.01$; ***$p < 0.001$; ****$p < 0.0001$).

The online version of this article includes the following source data and figure supplement(s) for figure 2:

**Source data 1.** Source data for *Figure 2C*.

**Source data 2.** Source data for *Figure 2C*.

**Figure supplement 1.** Histopathological analysis of intestines.

cells based on their co-expression of lineage marker genes (*Figure 4B and C*, *Figure 4—figure supplement 2A and B*). Due to severe epithelial disruption and inflammatory cell infiltration in *ctla-4*[-/-] zebrafish intestines, we focused on the pathological process and immune reactions in enterocytes and immune cell populations. KEGG analysis showed that apoptotic pathway was highly enriched in enterocytes of *ctla4*[-/-] zebrafish, suggesting that aberrant apoptosis contributes to the epithelial injury (*Figure 4—figure supplement 2C*). Subsequently, we conducted a TUNEL assay to detect apoptosis in the posterior intestines from both wild-type and *ctla4*[-/-] zebrafish. The results showed a higher number of apoptotic cells in the intestines of *ctla4*[-/-] zebrafish (*Figure 4—figure supplement 2D*). Additionally, genes functionally involved in the formation of tight and adhesion junctions, such as *oclna*, *cdh1*, *pcdh1b*, and *cldn15a*, were significantly down-regulated in enterocytes of *ctla-4*[-/-] zebrafish (*Figure 4D*), consistent with the pathological observation under electron microscope. Furthermore, inflammation-related genes and pathways were significantly up-regulated and enriched in neutrophils, B cells, and macrophages of *ctla-4*[-/-] zebrafish, suggesting active inflammatory responses (*Figure 4E–G*, *Figure 4—figure supplement 2E*). By sub-clustering analysis, six subpopulations were classified from T/NK/ILC-like cell groups based on their expression with a set of marker genes. These subpopulations include Cd8[+] T cells, ILC3-like cells, maturing Ccr7[high] T cells, NKT-like cells, and two groups of Th2 cells (*Figure 5A–C*, *Figure 5—figure supplement 1A*). The abundances of NKT-like and two subsets of Th2 cells were significantly increased in the intestines of *ctla-4*[-/-] zebrafish (*Figure 5D–F*). These findings were further validated by RT-qPCR detection of their corresponding marker genes (*Figure 5—figure supplement 1B and C*). These cells exhibited high expression levels of *il13* (*Figure 5G and H*). Specifically, the second subset of Th2 cells was seldom observed in the intestine of wild-type zebrafish, indicating their unique role in the pathogenesis of IBD-like phenotype in *ctla-4*[-/-] zebrafish (*Figure 5D–F*). KEGG analysis of up-regulated genes from *ctla-4*[-/-] NKT-like and Th2 cells indicated that Ctla-4 deficiency is positively associated with the inflammatory cytokine-cytokine receptor interaction, PPAR, calcium, and MAPK signaling pathways, cellular adhesion, and mucosal immune responses (*Figure 5I and J*, *Figure 5—figure supplement 1D*). Although the abundance of Cd8[+] T cells was not significantly changed in Ctla-4-deficient intestines, the inflammatory genes and pathways were up-regulated and enriched in the subset of T cells (*Figure 5—figure supplement 1E and F*). Notably, the proportion of ILC3-like cells was decreased, and they highly expressed *il17a/f1* and *il17a/f3* in the Ctla-4-deficient intestines (*Figure 5D–F and K*). Investigations have consistently reported a substantial decline in the population of ILC3s within the inflamed intestines and IL-17A-secreting ILC3s play a significant role in the development of intestinal inflammation (*Bernink et al., 2013*; *Li et al., 2017*; *Martin et al., 2019*; *Buonocore et al., 2010*; *Ermann et al., 2014*; *Aparicio-Domingo et al., 2015*). Thus, the reduced ILC3-like cells and increased expression of *il17a/f1* and *il17a/f3* may be responsible for intestinal inflammation induced by Ctla-4 deficiency.

## Decreased microbiota diversity in *ctla-4*[-/-] zebrafish intestines

The intestinal microbiota plays a crucial role in host functions such as nutrient acquisition, metabolism, epithelial cell development, and immunity. Notably, lower microbiota diversity has consistently been observed in patients with IBD phenotype (*Ott et al., 2004*; *Manichanh et al., 2006*), making it a valuable indicator of host health. Therefore, we further analyzed whether microbes are involved in the Ctla-4-deficiency induced intestinal inflammation. The results revealed a significantly higher number of amplicon sequence variants (ASVs) in wild-type zebrafish intestines, with 730 ASVs unique

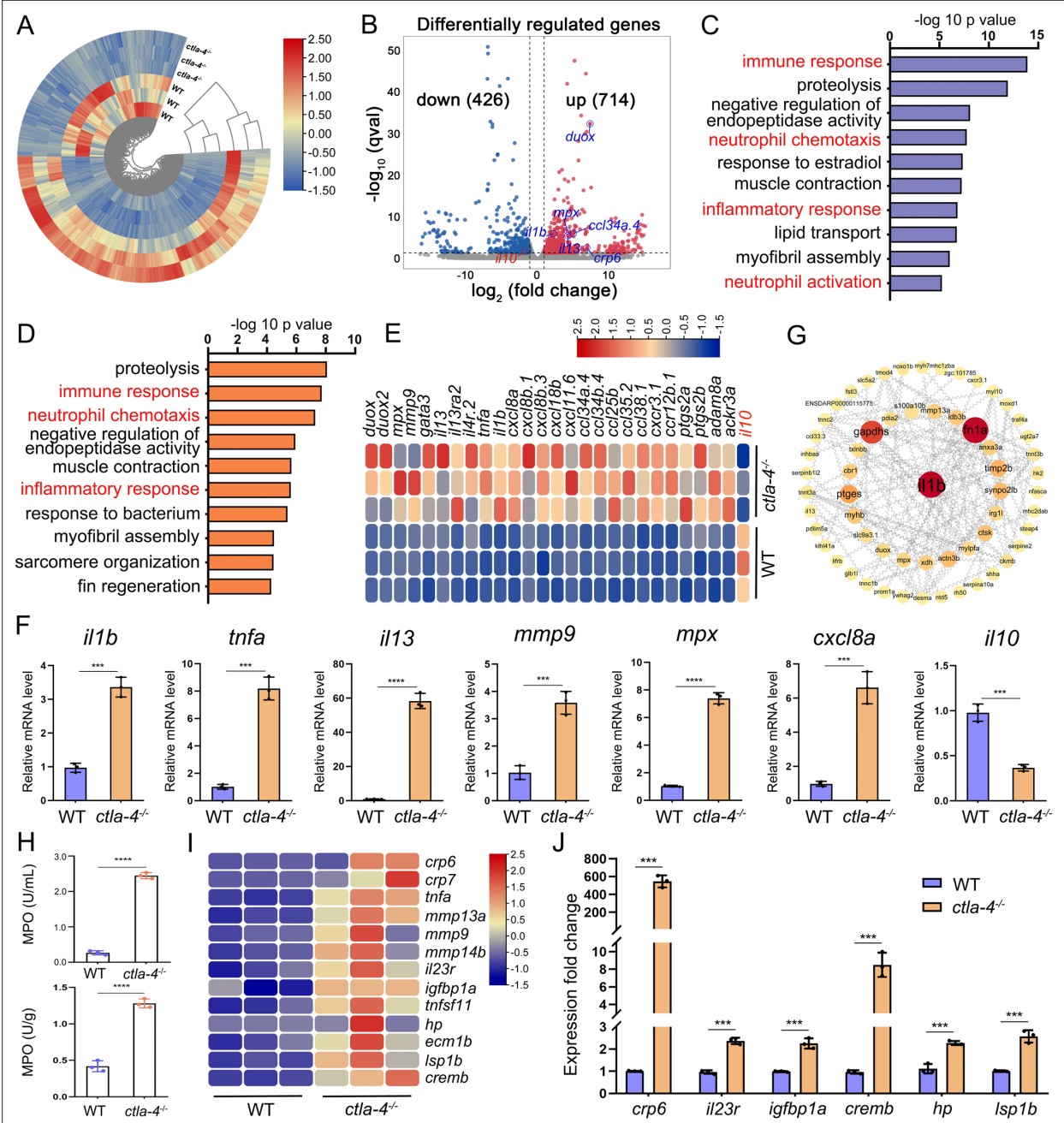

**Figure 3.** RNA-sequencing analysis of the molecular implications associated with the inflammatory bowel disease (IBD)-like phenotype in *ctla-4⁻/⁻* zebrafish. (**A**) Heatmap of differentially expressed genes between the intestines from wild-type (WT) and *ctla-4⁻/⁻* zebrafish. (**B**) Volcano plot showing the up-/down-regulated genes in the intestines of *ctla-4⁻/⁻* zebrafish compared with those of WT zebrafish. Red represents up-regulated genes, while blue denotes down-regulated genes. (**C**) GO analysis showing top 10 terms in biological processes of differentially expressed genes (DEGs). (**D**) GO analysis showing top 10 terms in biological processes of all up-regulated genes. (**E**) Heatmap showing row-scaled expression of the representative differently expressed inflammation and chemotaxis-related genes. (**F**) The mRNA expression levels of important genes associated with inflammation and chemokines confirmed by real-time qPCR. (**G**) Protein-protein interaction network was constructed using the DEGs. The nodes represent the proteins (genes); the edges represent the interaction of proteins (genes). (**H**) The myeloperoxidase (MPO) activity in the intestines (up) and peripheral blood (down). (**I**) Heatmap showing row-scaled expression of the IBD biomarker genes and IBD-related genes. (**J**) The mRNA expression levels of representative IBD biomarker genes and IBD-related genes were analyzed by real-time qPCR. Data are presented as mean ± standard deviation (SD). Statistical significance was assessed through an unpaired Student's t-test ($^{**}p < 0.01$; $^{***}p < 0.001$; $^{****}p < 0.0001$).

The online version of this article includes the following figure supplement(s) for figure 3:

**Figure supplement 1.** Examination on the functional genes and pathways associated with the inflammatory bowel disease (IBD)-like phenotype in *ctla-4⁻/⁻* zebrafish.

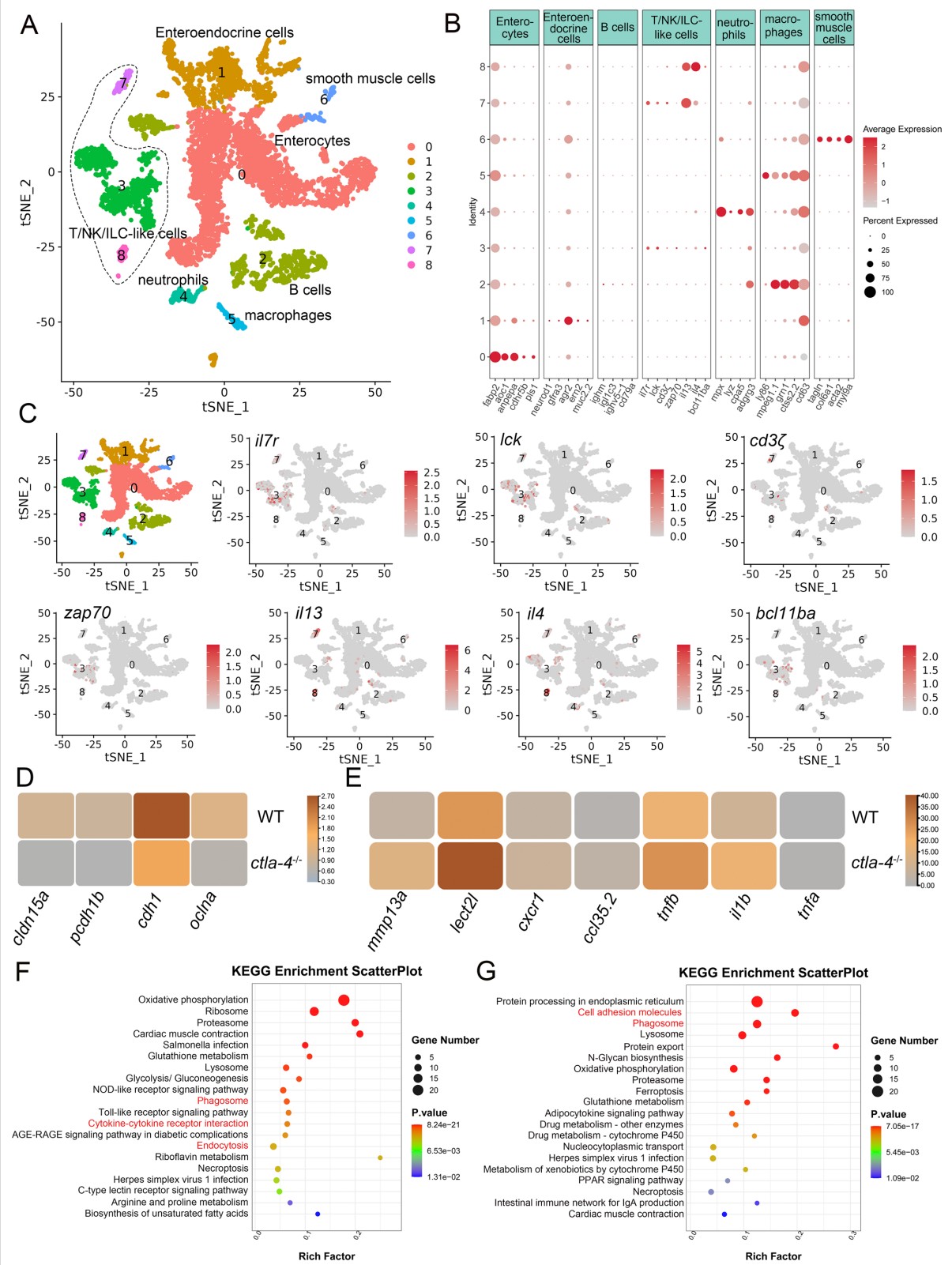

**Figure 4.** Single-cell RNA sequencing analysis of the major cell types associated with the inflammatory bowel disease (IBD)-like phenotype in *ctla-4⁻/⁻* zebrafish. (**A**) Classification of cell types from zebrafish intestines by tSNE embedding. (**B**) Dot plot showing the expression levels of lineage marker genes and percentage of cells per cluster that express the gene of interest. (**C**) Expression maps of T cell-associated markers within the cell populations of the zebrafish intestines. (**D**) Heatmap showing the mean expression levels of genes associated with tight and adhesion junctions in enterocytes across

*Figure 4 continued on next page*

*Figure 4 continued*

samples from wild-type (WT) and *ctla-4*⁻/⁻ zebrafish. (**E**) Heatmap showing the mean expression levels of inflammation-related genes involved in cytokine-cytokine receptor interactions in neutrophils from WT and *ctla-4*⁻/⁻ zebrafish samples. (**F**) KEGG enrichment analysis showing the top 15 terms of up-regulated genes in neutrophils from the *ctla-4*⁻/⁻ sample versus the WT sample. (**G**) KEGG enrichment analysis showing the top 15 terms of up-regulated genes in macrophages from the *ctla-4*⁻/⁻ sample versus the WT sample.

The online version of this article includes the following figure supplement(s) for figure 4:

**Figure supplement 1.** Quality control analysis of single-cell RNA sequencing data.

**Figure supplement 2.** Examination on the involvement of apoptotic process in epithelial cells and expression of inflammation-related genes in neutrophils and B cells in the intestines of *ctla-4*⁻/⁻ zebrafish.

to the wild-type group and 276 ASVs exclusively found in *ctla-4*⁻/⁻ group (*Figure 6A*). Furthermore, the Shannon index and the Simpson index indicated a decreased microbial diversity in *ctla-4*⁻/⁻ zebrafish intestines (*Figure 6B*) and the Principal Coordinate Analysis (PCoA) using Bray-Curtis distance revealed a significant separation in the microbial composition between *ctla-4*⁻/⁻ group and the wild-type group (*Figure 6C*). To gain insights into the microbial community composition, we analyzed the identified microbial populations at the class and family level. Alphaproteobacteria and Gammaproteobacteria were found to be the most prevalent classes. Relative to wild-type group, Ctla-4 deficiency resulted in a significant reduction in Alphaproteobacteria abundance. However, the Gammaproteobacteria, one of the main classes of Gamma-negative pathogenic bacteria expanded under inflammation conditions, was increased, although the change did not reach statistical significance (*Figure 6D and E*; *Zhao et al., 2023*). In addition, we observed a decreased relative abundance of short-chain fatty acids (SCFAs)-producing Bacilli and Verrucomicrobiae, the latter of which contributes to glucose homeostasis and intestinal health (*Figure 6F and G*; *Xu et al., 2020*; *Belzer and de Vos, 2012*). Notably, the family-level analysis revealed a notable enrichment of Enterobacteriaceae, overgrowing under host inflammatory conditions, and the Shewanellaceae, serving as the most important secondary or opportunistic pathogens, in *ctla-4*⁻/⁻ zebrafish (*Figure 6H and I*). To identify differentially abundant bacterial taxa between the wild-type and *ctla-4*⁻/⁻ zebrafish, we conducted linear discriminant analysis (LDA) effect size (LEfSe). The results showed that several potentially opportunistic pathogens, including Gammaproteobacteria, Enterobacterales, and Aeromonadales were found to be overrepresented in *ctla-4*⁻/⁻ zebrafish (*Figure 6J*). In contrast, Actinobacteriota, Cetobacterium, and Planctomycetota (Planctomycetes) were more abundant in the wild-type zebrafish. These findings strongly indicated an association between Ctla-4 deficiency-induced gut inflammation and dysbiosis, as characterized by decreased microbial diversity, loss of potentially beneficial bacteria, and expansion of pathobionts.

## Inhibitory role of Ctla-4 in T cell activation

Given that Ctla-4 is primarily expressed on T cells (*Figure 1E–F*), its absence has been shown to induce intestinal immune dysregulation, indicating a crucial role of this molecule as a conserved immune checkpoint in T cell inhibition. Mechanistically, Ctla-4 may inhibit T cell activation by obstructing the Cd80/86-Cd28 costimulatory pathway, a mechanism conserved in mammalian species. To elucidate the regulatory role of Ctla-4 in costimulatory signal-dependent T cell activation, we conducted a series of blockade and activation assays using anti-Ctla-4 antibody, recombinant soluble Ctla-4-Ig (sCtla-4-Ig), sCd28-Ig, and sCd80/86 proteins in a PHA-stimulating and mixed lymphocyte reaction (MLR) model. In this system, sCtla-4-Ig and sCd28-Ig served as antagonists to block membrane-bound Cd80/86, while sCd80/86 acted as an agonist for membrane-bound Cd28 (*Figure 7—figure supplement 1A–C*). As expected, the proliferation of lymphocytes from *ctla-4*⁻/⁻ zebrafish was more pronounced compared to wild-type controls, and the addition of sCtla-4-Ig effectively suppressed this proliferation (*Figure 7A and B*). These findings indicate that the absence of Ctla-4 leads to enhanced lymphocyte activation, which can be counteracted by sCtla-4 administration, underscoring the inhibitory function of Ctla-4 in T cell regulation. Consistent with these results, blockade of Ctla-4 using anti-Ctla-4 Ab significantly promoted the proliferation of lymphocytes from wild-type zebrafish (*Figure 7C*). Furthermore, sCd28-Ig administration inhibited the proliferation of lymphocytes from *ctla-4*⁻/⁻ zebrafish (*Figure 7D*), whereas sCd80/86 promoted the expansion of Ctla-4-deficient lymphocytes (*Figure 7E*). Based on these results, we concluded that the presence of Ctla-4 obstructs the Cd80/86-Cd28-mediated costimulatory signaling, consequently impeding cell proliferation. To further investigate the molecular interactions between Cd28, Ctla-4, and Cd80/86, we employed AlphaFold2

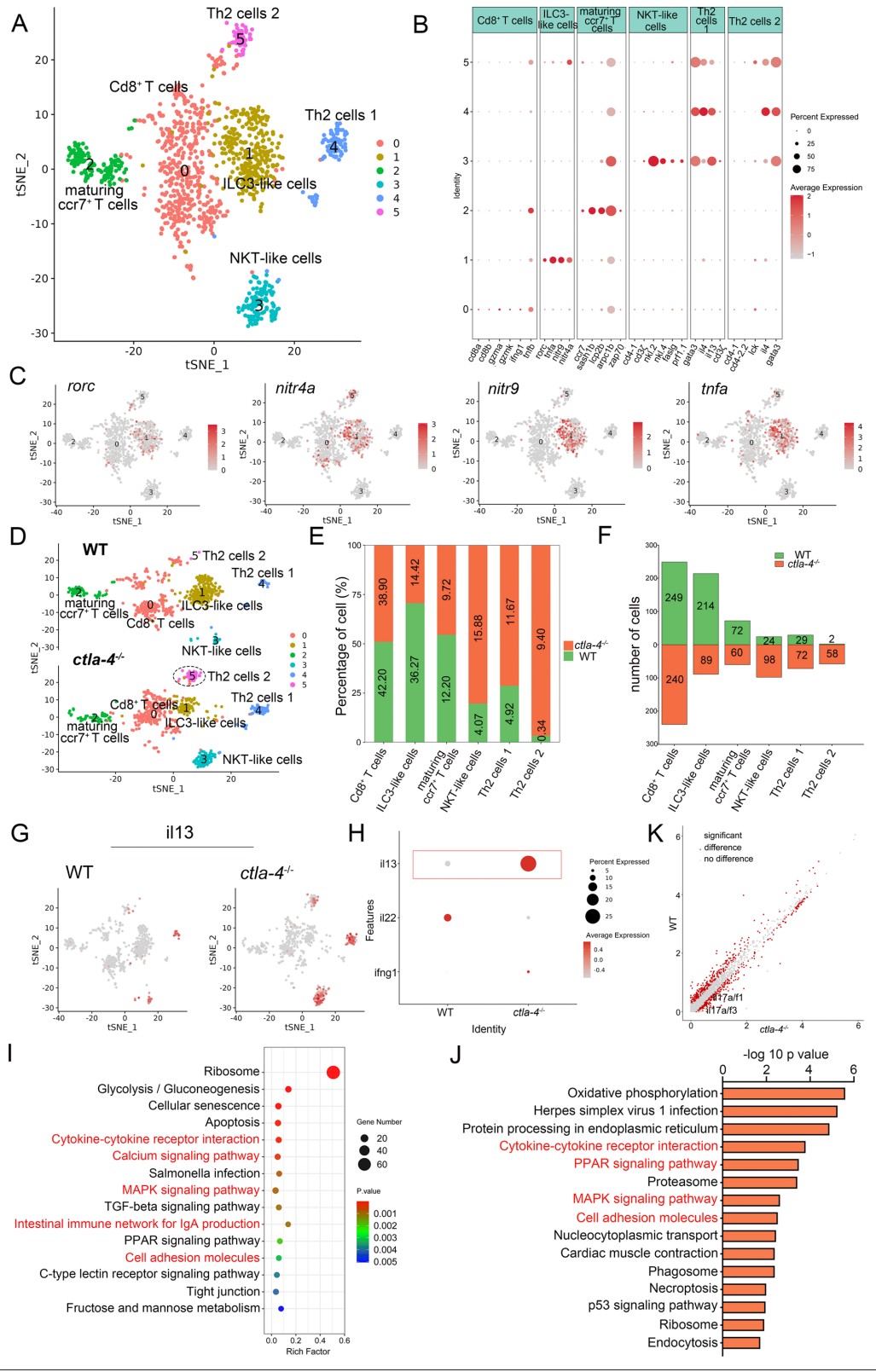

**Figure 5.** Single-cell RNA sequencing analysis of the subset of immune cells associated with the inflammatory bowel disease (IBD)-like phenotype in *ctla-4*-/- zebrafish. (**A**) Classification of subset cells from the T/NK/ILC-like group by tSNE embedding. (**B**) Dot plot showing the mean expression levels of subset marker genes and percentage of cells per cluster that express the gene of interest. (**C**) Marker gene expression in individual cluster

*Figure 5 continued on next page*

*Figure 5 continued*

identifying this cluster as ILC3-like cells. (**D**) Changes in the composition of subset cells between samples from wild-type (WT) and *ctla-4⁻/⁻* zebrafish. A significantly increased Th2 subset (referred to as Th2 cells 2) in the *ctla-4⁻/⁻* sample was highlighted with a black dashed circle. (**E**) Histogram showing the different ratios of subset cells between the WT and *ctla-4⁻/⁻* samples. (**F**) Histogram presenting the different numbers of subset cells between the WT and *ctla-4⁻/⁻* samples. (**G**) Mean expression levels of the cytokine *il13* within different subset cells between the WT and *ctla-4⁻/⁻* samples. (**H**) Dot plot illustrating the mean expression of *il13* in T/NK/ILC-like cells from WT and *ctla-4⁻/⁻* zebrafish. (**I**) KEGG enrichment analysis showing the top 15 terms of the Th2 cells 2 genes from *ctla-4⁻/⁻* zebrafish. (**J**) KEGG enrichment analysis showing the top 15 terms of up-regulated genes in NKT-like cells. (**K**) Scatter plot showing the differentially expressed genes (DEGs) of ILC3-like cells in WT and *ctla-4⁻/⁻* zebrafish. The *il17a/f1* and *il17a/f3* was shown in the scatter plot.

The online version of this article includes the following figure supplement(s) for figure 5:

**Figure supplement 1.** Examination on the activation of T cell subsets in the intestines of *ctla-4⁻/⁻* zebrafish.

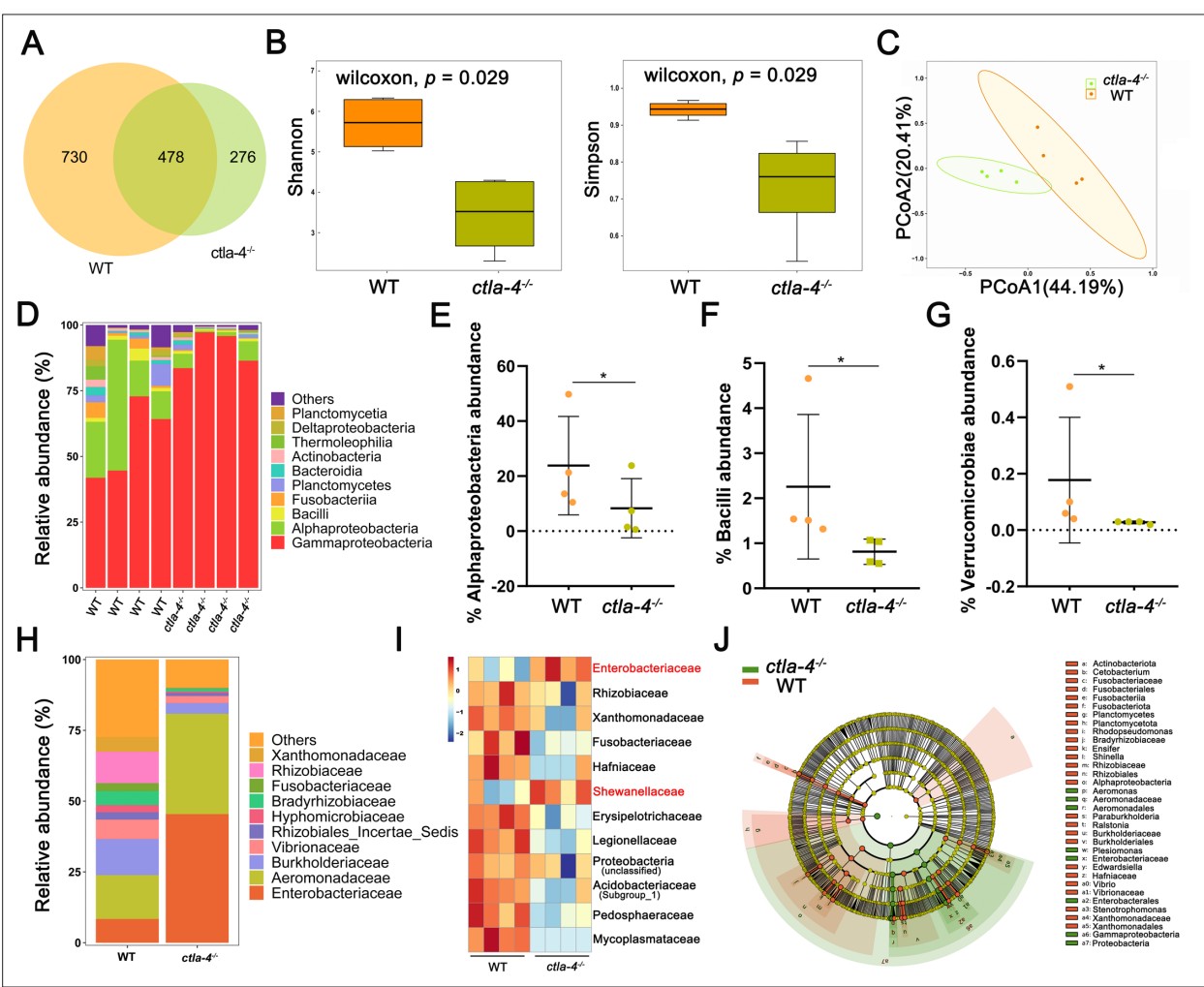

**Figure 6.** Alteration in microbial composition in the intestines of *ctla-4⁻/⁻* zebrafish. (**A**) Venn diagram showing the number of amplicon sequence variants (ASVs) in zebrafish intestinal microbiota. (**B**) Alpha diversity of microbes was calculated through Shannon index and Simpson index. (**C**) Beta-diversity analyzed based on Principal Coordinate Analysis (PCoA) was shown by using Bray-Curtis distance. (**D**) The relative abundance of intestinal microbiota at the class level. (**E–G**) The relative abundance of Alphaproteobacteria (**E**), Bacilli (**F**), and verrucomicrobiae (**G**) in the intestines from the wild-type (WT) and *ctla-4⁻/⁻* zebrafish. $^*p < 0.05$. (**H**) The relative abundance of intestinal microbiota at the family level. (**I**) Heatmap showing row-scaled expression of the differential abundances of bacterial communities at family level in the WT and *ctla-4⁻/⁻* zebrafish ($p < 0.05$). (**J**) Cladogram representation of LEfSe analysis showing the differentially abundant bacterial taxa between the intestines from WT (red) and *ctla-4⁻/⁻* (green) zebrafish ($p < 0.05$).

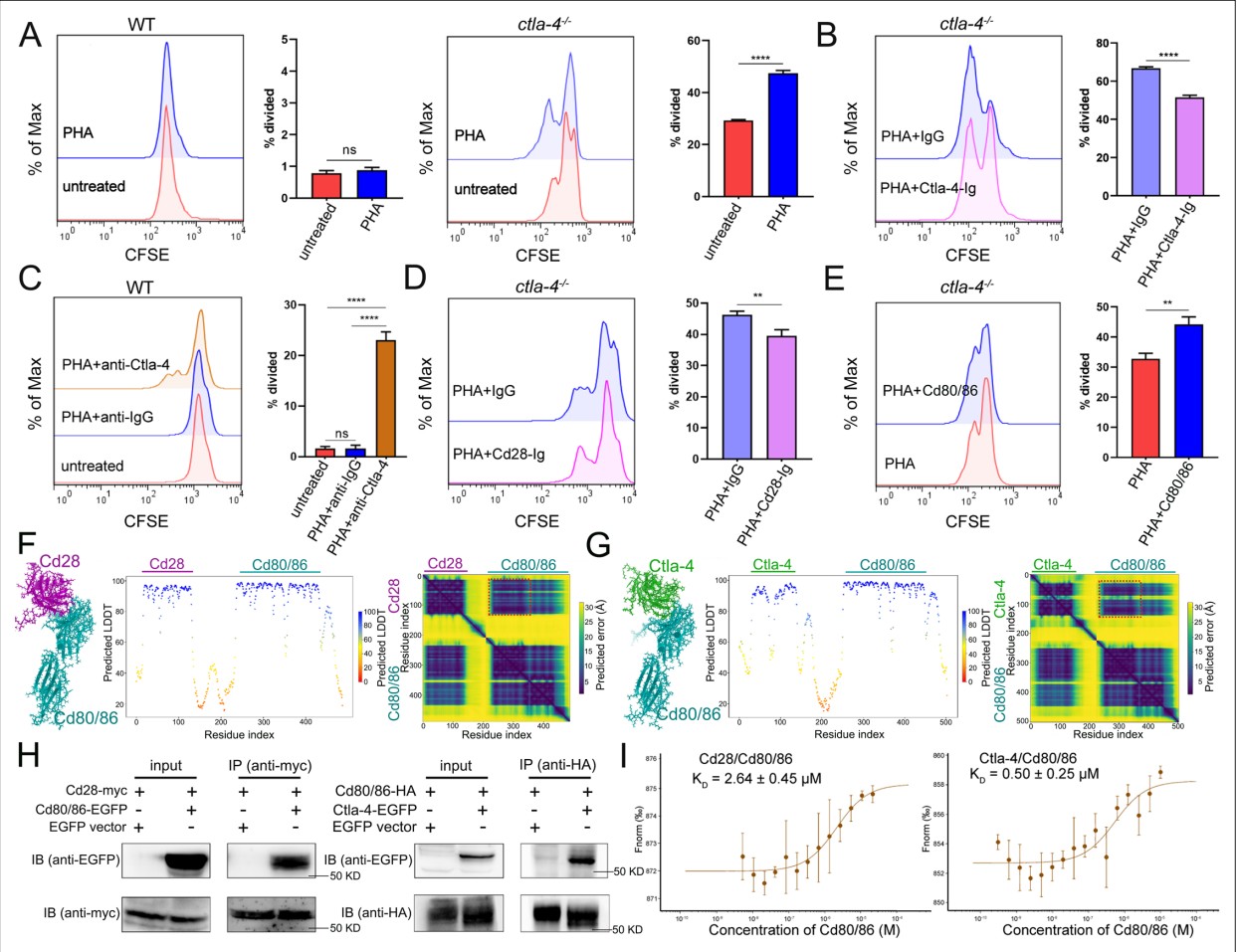

**Figure 7.** Examination on the inhibitory function of Cytotoxic T lymphocyte antigen-4 (Ctla-4) in T cell activation. (**A**) Assessment of the proliferative activity of T cells from wild-type (WT) and *ctla-4*[-/-] zebrafish by a mixed lymphocyte reaction combined with PHA-stimulation. The carboxyfluorescein succinimidyl ester (CFSE) dilution, which served as an indicator of lymphocyte proliferation, was measured through flow cytometry. (**B**) Assessment of the proliferative activity of lymphocytes from *ctla-4*[-/-] zebrafish by the administration of sCtla-4-Ig. (**C**) Assessment of the proliferative activity of lymphocytes from WT zebrafish by supplementing anti-Ctla-4 antibody. (**D**) Assessment of the proliferative activity of lymphocytes from *ctla-4*[-/-] zebrafish by the administration of sCd28-4-Ig. (**E**) Assessment of the proliferative activity of lymphocytes from *ctla-4*[-/-] zebrafish by the administration of recombinant sCd80/86 protein. (**F, G**) Interactions between Cd80/86 and Cd28 (**F**), and Cd80/86 and Ctla-4 (**G**) as predicted by AlphaFold2. On the left are structural models depicting Cd80/86 in complex with Cd28 or Ctla-4. The center panels display per-residue model confidence scores (pLDDT) for each structure, using a color gradient from 0 to 100, where higher scores indicate increased confidence. The right panels show the predicted aligned error (PAE) scores for each model. The well-defined interfaces between Cd28 or Ctla-4 and Cd80/86 are highlighted with red dashed squares. (**H**) The interaction between Cd80/86 and Cd28 (left), and Cd80/86 and Ctla-4 (right) were verified by Co-IP. (**I**) Binding affinities of the Cd80/86 protein for the Cd28 and Ctla-4 proteins were measured by the microscale thermophoresis (MST) assay. The $K_D$ values are provided. Data are presented as mean ± standard deviation (SD), which were calculated from three independent experiments. Statistical significance was assessed through an unpaired Student's t-test (**p < 0.01; ***p < 0.001; ns denotes no statistical significance).

The online version of this article includes the following source data and figure supplement(s) for figure 7:

**Source data 1.** Source data for *Figure 7H*.

**Source data 2.** Source data for *Figure 7H*.

**Figure supplement 1.** Preparation of recombinant proteins and examination of their molecular interactions.

**Figure supplement 1—source data 1.** Source data for *Figure 7—figure supplement 1A–C*.

**Figure supplement 1—source data 2.** Source data for *Figure 7—figure supplement 1A–C*.

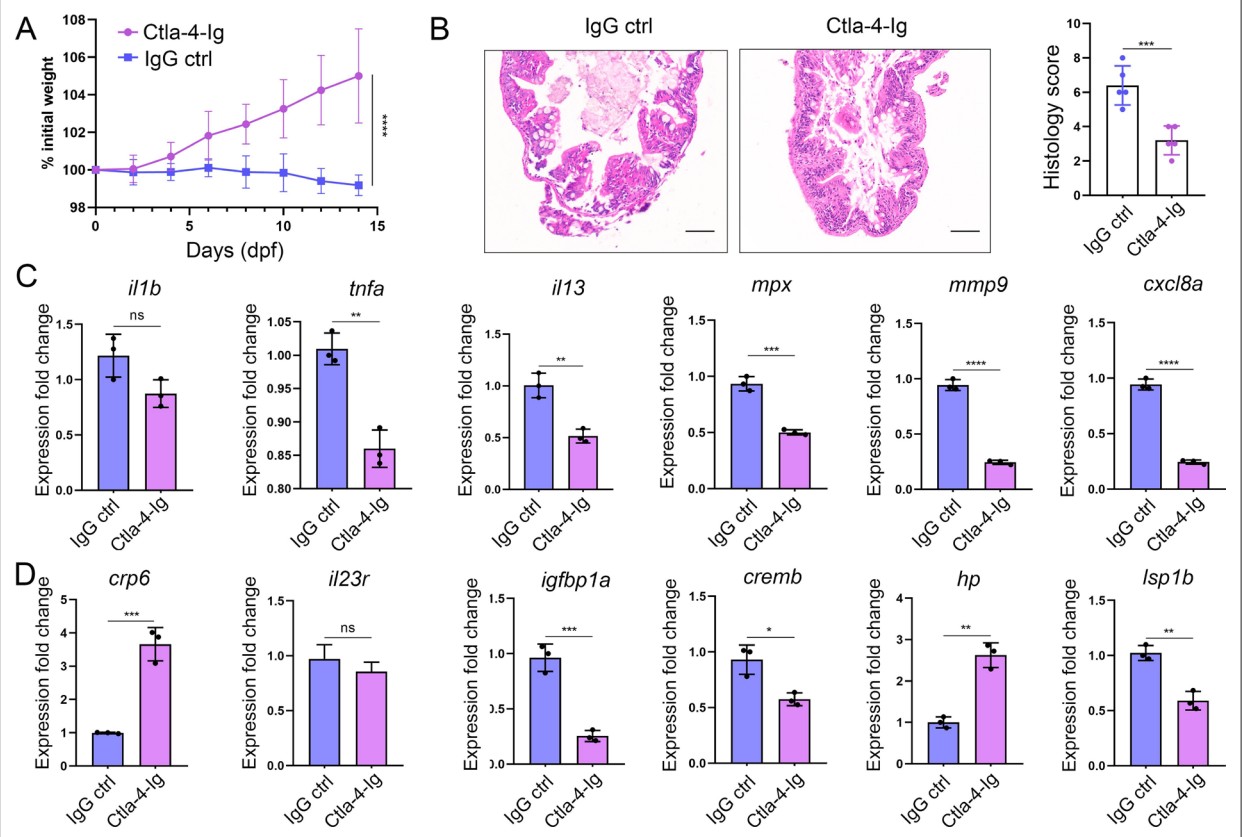

**Figure 8.** In vivo inhibition of intestinal inflammation by sCtla-4-Ig. (**A**) Percent initial weight of zebrafish after injection of the sCtla-4-Ig or the IgG isotype control. Each group consisted of six zebrafish (n=6). Data show means with SEM analyzed by two-way ANOVA with Sidak's correction for multiple comparisons. (**B**) Representative H&E staining analysis of histopathological changes and quantitation of histology scores in the posterior intestine from *ctla-4⁻/⁻* zebrafish treated with sCtla-4-Ig or IgG isotype control. Scale bar: 50 μm. (**C**) The mRNA expression levels of inflammation-related genes in *ctla-4⁻/⁻* zebrafish treated with sCtla-4-Ig or IgG isotype control. (**D**) The mRNA expression levels of IBD biomarker genes and IBD-related genes in *ctla-4⁻/⁻* zebrafish treated with sCtla-4-Ig or IgG isotype control. The p-value was generated by an unpaired two-tailed Student's t-test. **$p < 0.01$; ***$p < 0.001$; ****$p < 0.0001$.

to predict the structures of Cd80/86-Cd28 and Cd80/86-Ctla-4 complexes. A total of 25 models were generated for each complex and subsequently aligned with Cd80/86. The predictions indicated that both Cd28 and Ctla-4 form a well-defined interface with Cd80/86, utilizing the same binding site (*Figure 7—figure supplement 1D and E*). This well-defined interface was corroborated by lower predicted aligned error (PAE) scores for each model, as marked by the red dashed square (*Figure 7F and G*). Subsequently, co-immunoprecipitation (Co-IP) assays were conducted to provide compelling evidence for the molecular interactions between Cd80/86 and Cd28 or Ctla-4. Flow cytometry analysis further revealed dose-dependent associations between Cd80/86 and Cd28 or Ctla-4 in HEK293T cells (*Figure 7H*, *Figure 7—figure supplement 1F*). Additionally, microscale thermophoresis assays demonstrated that Ctla-4 exhibits a higher binding affinity for Cd80/86 than Cd28, as evidenced by a lower equilibrium association constant value ($K_D = 0.50 \pm 0.25$ μM vs. $K_D = 2.64 \pm 0.45$ μM) (*Figure 7I*). These findings suggest that Ctla-4 exerts its inhibitory function by competing with Cd28 for binding Cd80/86.

## sCtla-4-Ig alleviates IBD-like phenotype

As described above, engagement of Cd80/86 by sCtla-4-Ig effectively suppressed T cell activation in vitro (*Figure 7B*), indicating that sCtla-4-Ig holds promise as a potential intervention for IBD-like phenotype. This is supported by the observation that Ctla-4-deficient zebrafish treated with sCtla-4-Ig exhibited obvious body weight restoration compared to those treated with the IgG isotype control (*Figure 8A*). To provide further evidence, histological analysis was performed on the posterior

intestine, which is known to experience severe tissue destruction under Ctla-4 deficient conditions. As expected, Ctla-4-Ig treatment resulted in a significant decrease in lymphocyte infiltration and an improvement in the epithelial barrier (*Figure 8B*). Moreover, Ctla-4-Ig treatment significantly reduced the expression of pro-inflammatory genes, including *il13*, *tnfa*, *mpx*, *mmp9*, and *cxcl8a*, as well as *igfbp1a*, *cremb*, and *lsp1a*, which are susceptibility genes for IBD observed in mammals (*Figure 8C and D*). These findings demonstrate that the supplementation of Ctla-4-Ig alleviates intestinal inflammation in Ctla-4-deficient zebrafish, highlighting its potential as a therapeutic intervention for CTLA-4 deficiency-induced IBD in mammals.

## Discussion

As an essential negative regulator of T cell activation, dysfunction of CTLA-4 was implicated in various diseases in both humans and murine models (*Tivol et al., 1995*; *Waterhouse et al., 1995*; *Hosseini et al., 2020*). Numerous previous studies have established the connection between CTLA-4 and autoimmune thyroiditis, Graves' disease, myocarditis, pancreatitis, multiple sclerosis, rheumatoid arthritis, and type I diabetes (*Sun et al., 2019*; *Vergara et al., 2024*; *Khalid Kheiralla, 2021*; *Lin et al., 2022*; *Cutolo et al., 2016*; *Chang et al., 2007*; *Fathima et al., 2019*). However, the involvement of CTLA-4 in IBD has been understudied. Several investigations have reported that haploinsufficiency resulting from mutations in CTLA-4 in humans is associated with IBD, and genome-wide association studies (GWAS) have implicated CTLA-4 as a susceptibility gene for IBD (*Zeissig et al., 2015*; *Angelino et al., 2021*; *Liu et al., 2015*). Nevertheless, the exact contributions and mechanisms of CTLA-4 deficiency in the occurrence and pathology of IBD remain incompletely understood, primarily due to the lack of animal models attributable to the lethality of CTLA-4 knockout in mice. In this study, we identified the Ctla-4 homolog in zebrafish, and discovered that defects in Ctla-4 did not have a severe lethal effect, but did show a clear IBD-like phenotype. This makes zebrafish an attractive animal model for investigating the molecular and cellular mechanisms underlying Ctla-4-mediated IBD.

Multiple lines of histopathological evidence demonstrated the IBD-like phenotype in Ctla-4-deficient zebrafish. Key features include epithelial hyperplasia, disruption of epithelial integrity, loss of goblet cells, increased acidic mucus production, intestinal lumen enlargement, inflammatory cell infiltration, and elevated expression of pro-inflammatory cytokines in the inflamed intestines. These characteristics, such as epithelial hyperplasia, goblet cell depletion, inflammatory cell infiltration, and upregulated pro-inflammatory cytokine expression, closely resemble those observed in the dextran sodium sulfate (DSS)-induced IBD model in mice (*Kim et al., 2012*). Similarly, mononuclear cell infiltration and significant upregulation of the il1β cytokine have been reported in the trinitrobenzenesulfonic acid (TNBS)-induced IBD model in adult zebrafish (*Geiger et al., 2013*). In zebrafish larvae, the TNBS-induced IBD-like phenotype also exhibits an enlarged intestinal lumen, although goblet cell numbers were increased (*Fleming et al., 2010*). Additionally, neutrophilic inflammation and acidic mucin accumulation have been observed in the DSS-induced enterocolitis model in zebrafish larvae (*Oehlers et al., 2012*). In contrast, the soybean meal-induced enteritis (SBMIE) phenotype in zebrafish larvae shows no significant structural changes in intestinal architecture, despite an increased number of neutrophils and lymphocytes (*Coronado et al., 2019*). In summary, Ctla-4 deficiency induces IBD-like phenotypes analogous to those typically elicited by drugs in mice and zebrafish, making this model a valuable tool for comprehending the pathophysiological mechanisms underlying IBD.

A transcriptomics study was conducted to investigate the mechanisms of Ctla-4-deficiency induced IBD. RNA-seq analysis demonstrated a significant upregulation of important inflammatory cytokines, such as *il1b* and *tnfa* in the Ctla-4-deficient intestines. This is consistent with studies showing that IL-1β and TNF-α act as crucial mediators in mammalian IBD models by disrupting epithelial junctions and inducing apoptosis of epithelial cells (*Al-Sadi et al., 2013a*; *Al-Sadi et al., 2013b*). Conversely, the key anti-inflammatory cytokines, such as *il10*, were downregulated. These findings highlight an imbalance between pro-inflammatory and anti-inflammatory cytokines in the intestines of Ctla-4-deficient fish. Consistently, the inflammatory signaling pathways associated with these upregulated cytokines, such as the ERK1/2 and MAPK pathways, were positively enriched in inflamed intestines. Single-cell RNA-seq analysis revealed the upregulation and enrichment of these inflammatory cytokines and pathways in neutrophils, macrophages, and B cells of inflamed intestines, indicating their active involvement in inflammatory responses and as major sources of inflammatory signals. Additionally, there was a marked increase in the abundance of Th2 subset cells in the inflamed intestines. These cells exhibited

high expression of *il13* and were significantly enriched in inflammatory signaling pathways, indicating their activated state. These findings align with previous studies indicating that T cells in CTLA-4-deficient mice exhibit a bias toward Th2 differentiation (*Khattri et al., 1999*; *Bour-Jordan et al., 2003*). Furthermore, IL-13, a key effector Th2 cytokine, has been implicated in the pathogenesis of ulcerative colitis in mammals, where it directly contributes to epithelial cell damage by disrupting tight junctions, inducing apoptosis, and impairing cellular restitution (*Heller et al., 2005*). Therefore, upregulated Il13 from Th2 cells may be a significant contributor to the occurrence of intestine inflammation in Ctla-4-deficient zebrafish. Notably, the proportion of ILC3-like cells was downregulated in the inflamed intestines, consistent with recent studies reporting a substantial decline of ILC3 in IBD patients (*Bernink et al., 2013*; *Li et al., 2017*; *Martin et al., 2019*). ILC3 is the most abundant type of ILCs in the intestines and plays a protective role in IBD in mammals by promoting epithelial cell proliferation and survival, as well as enhancing intestinal barrier function through the production of IL-22 (*Buonocore et al., 2010*; *Aparicio-Domingo et al., 2015*). Thus, the marked decrease in ILC3-like cells may exacerbate intestinal inflammation and damage.

IBD is frequently associated with alterations in gut microbiota composition, characterized by reduced microbial diversity and an imbalance between beneficial and pathogenic bacteria. The common feature of these changes is the expansion of Proteobacteria, particularly members of the Enterobacteriaceae family (*Shin et al., 2015*; *Winter et al., 2013*). Similarly, Ctla-4-deficient zebrafish exhibited significant enrichment of Enterobacteriaceae, alongside a decline in beneficial bacteria like Cetobacterium and an increase in opportunistic pathogens such as γ-Proteobacteria and Aeromonadales. These findings indicate shared patterns in microbial flora changes during intestinal inflammation. Previous studies suggest that reduced microbial diversity in IBD results from the loss of normal anaerobic bacteria, such as Bacteroides, Eubacterium, and Lactobacillus (*Ott et al., 2004*). Concurrently, inflammation-driven increases in intestinal lumen oxygenation and the availability of nitrate and host-derived electron acceptors facilitate anaerobic respiration and Enterobacteriaceae proliferation (*Hughes et al., 2017*). These observations highlight the intricate interplay between IBD pathogenesis, gut microbial alterations, and host immune homeostasis. The zebrafish IBD-like model induced by Ctla-4 deficiency offers new insights into this research area. For instance, abnormal activation of Th2 cells may lead to dysfunction in downstream B cells and mucosa-associated immunity, which are crucial for maintaining symbiotic bacterial homeostasis in the intestines (*Xu et al., 2020*). This suggests a potential link between Th2 cell changes and the observed alterations in the intestinal microbial community in Ctla-4-deficient zebrafish. Moreover, Ctla-4 deficiency alters the proportion and activation of ILC3 cells and damages the intestinal epithelium, potentially shaping the inflammatory milieu and further disrupting gut microbial homeostasis. Ctla-4 also regulates T cell activation by inhibiting the Cd80/86 co-stimulatory pathway. These findings suggest a regulatory interplay between Ctla-4, ILC3 cells, Cd80/86-primed T cells, and gut microbiota in Ctla-4 deficiency-induced IBD. Recently, gut microbiota exposure has been found to induce local IL-23 production, which upregulates CTLA-4 on ILC3s. This supports immune regulation by reducing CD80/86 co-stimulatory signaling and increasing PD-L1 bioavailability on myeloid cells. Impairment of this pathway manifests in a substantial imbalance of effector and regulatory T cell responses, exacerbating intestinal inflammation (*Ahmed et al., 2024*). These findings bolster our hypothesis and provide valuable insights into the complex interactions between gut microbiota, ILC3-mediated immune responses, and Cd80/86 signaling in Ctla-4 deficiency-induced IBD.

In conclusion, our study demonstrates that Ctla-4 serves as a potential genetic determinant of the IBD-like phenotype in zebrafish, although further research is necessary to conclusively identify the causative variant responsible for this association. The development of this zebrafish model offers a valuable tool for elucidating the mechanisms underlying the disease's pathophysiology. Nevertheless, a deeper understanding of the intricate interactions among immune cells, intestinal epithelial cells, and the microbiome in IBD remains an area warranting further investigation.

## Materials and methods
### Experimental fish
The AB strain zebrafish (*Danio rerio*) of both sexes, 4–6 mo of age with body weights ranging from 0.3 to 0.8 g and lengths of 3–4 cm, were reared in the laboratory in recirculating water at 26–28°C

under standard conditions as previously described (*Shi et al., 2019*). All animal experiments were performed in compliance with legal regulations and approved by the Research Ethics Committee of Zhejiang University. For sampling, wild-type and Ctla-4-deficient zebrafish of varying ages were kept in separate tanks and labeled with their respective dates of birth. Wild-type zebrafish aged 4–6 mo and Ctla-4-deficient zebrafish aged 4 mo were used for the experiments.

## Generation of Ctla-4-deficient zebrafish

CRISPR/Cas9 system was used to knock out the *ctla-4* gene. The targeting sequence 5'-CTCAGAGC CCTACTTCGCAA-3' was designed by optimized CRISPR Design (http://crispr.mit.edu/) and synthesized by T7 RNA polymerase and purified by MEGAclear Kit (AM1908; Invitrogen) in vitro. Cas9 protein (500 ng/µl, A45220P; Thermo Fisher Scientific) and purified RNA (90 ng/µl) were coinjected into one cell-stage wild-type embryos. For genotyping, DNA fragment was amplified with primers (F: 5' -TGTGACAGGAAAAGATGGAGAA- 3' and R: 5'- GATCAGATCCACTCCTCCAAAG- 3') at 94°C for 4 min followed by 35 cycles at 94°C for 30 s, 58°C for 30 s, and 72°C for 30 s, culminating in a final extension at 72°C for 10 min. Subsequently, the PCR product was subjected to sequencing. The mutant alleles (–14 bp) were obtained. As with wild-type zebrafish, Ctla-4-deficient zebrafish were reared in the laboratory in recirculating water at 26–28°C under standard conditions.

## Preparation of recombinant proteins

For prokaryotic expression, the encoding sequences for the extracellular domains of Ctla-4 and Cd80/86 (designated as soluble Cd80/86, sCd80/86) were amplified and cloned into the pET-28a (+) and pCold-GST vectors. The primers used are shown in *Supplementary file 1*. The recombinant plasmid was transformed into *Escherichia coli* BL21 (DE3) competent cells (TransGen Biotech) and induced with isopropyl-β-D-thiogalactoside (IPTG, 0.5 mM) at 20°C for 12 hr. After ultrasonication, the supernatants were collected for purification. For eukaryotic Ctla-4-Ig and Cd28-Ig expression, the extracellular domains of Ctla-4 and Cd28 were fused to the Fc region of human IgG1 (*Linsley et al., 1991*), and cloned into the pAcGHLTc vector. The recombinant plasmids were transfected into sf9 (*Spodoptera frugiperda*) cells with baculovirus vector DNA (AB Vector) under the assistance of lipofectamine 3000 (Thermo Fisher Scientific). The cells were cultured at 28°C for 3 d and subsequently dissolved in a lysing buffer (50 mM Tris-HCl, pH 8.0, 150 mM NaCl, 1% Nonidet P-40, 1 mM PMSF). The recombinant Ctla-4-Ig and Cd28-Ig proteins (designated as soluble Ctla-4-Ig, sCtla-4-Ig, and sCd28-Ig) were purified using Ni-NTA agarose affinity chromatography (QIAGEN) following the manufacturer's protocol. Proteins were then separated on a 12% SDS-PAGE gel and visualized through Coomassie Brilliant Blue R250 staining.

## Preparation of polyclonal antibody

Antibody (Ab) against the Ctla-4 extracellular domain protein was produced through a recombinant protein immunized approach as previously described (*Shi et al., 2019*). Briefly, 4-wk-old male BALB/c mice (~15 g) were immunized with recombinant Ctla-4 protein with extracellular domain (25 µg) each time in CFA (Sigma-Aldrich) initially and then in IFA (Sigma-Aldrich) for four times thereafter at biweekly intervals. Seven days after the final immunization, serum samples were collected when antiserum titers exceeded 1:10,000. The Ab was affinity purified by Protein-A Agarose Columns (Thermo Fisher Scientific), and its titer was examined by ELISA. The validity and specificity of the Ab was determined by Western blot analysis.

## Subcellular localization

HEK293T cells were seeded into the 12-well plates (Corning) with cover glass and cultured in high-glucose DMEM (Gibco) supplemented with 10% FBS (Cell-Box) at 37°C in 5% $CO_2$ to allow growth until 40–50% confluence. The cells were transfected with pEGFPN1-Ctla-4 plasmid DNA (0.8 µg) with the help of PEI reagent (3.2 µg per well) in accordance with the manufacturer's protocol. After transfection for 24 hr, the cells were fixed with 4% paraformaldehyde (PFA; Sigma-Aldrich) and stained with CM-DiI (1 µM; Thermo Fisher Scientific) and DAPI (100 ng/ml; Sigma-Aldrich). The fluorescence images were obtained using a two-photon laser scanning confocal microscope (LSM710; Zeiss, Jena, Germany) with 630x magnification.

## Identification of monomer or dimer

HEK293T cells were transfected with pCDNA3.1-HA-Ctla-4 (0.8 µg) or pCDNA3.1-HA (0.8 µg) under the assistance of polyethylenimine (PEI; Sigma-Aldrich). After 48 hr, the cells were lysed with precooling cell lysis buffer (Beyotime) and the supernatants were mixed with non-reducing (without β-ME) or reducing (with β-ME) loading buffer for Western blot analysis.

## Immunofluorescence staining

Colocalization of Cd4-1/Cd8α and Ctla-4 was determined by immunofluorescence staining. Leukocytes were isolated from zebrafish's spleen, kidney, and peripheral blood by Ficoll-Hypaque (1.080 g/ml; Sangon Biotech) centrifugation at 2500 rpm at 25°C for 25 min. After washing with D-Hank's solution, cells were fixed with 4% PFA at room temperature for 10 min, blocked with 2% BSA (Sigma-Aldrich), and incubated with primary Abs at 4 °C for 2 hr. The primary Abs included combinations of rabbit anti-CD4-1 and mouse anti-CTLA-4, or rabbit anti-CD8α and mouse anti-CTLA-4, which were produced in our laboratory as previously described (*Shi et al., 2019*). Following another wash with D-Hank's solution, the cells were combined with secondary Abs, including FITC-conjugated goat anti-rabbit IgG and PE-conjugated goat anti-mouse IgG (Thermo Fisher Scientific), according to the manufacturer's instructions. After a final wash with D-Hank's solution, the cells were stained with DAPI (100 ng/ml) at room temperature for 10 min. Fluorescence images were captured using a two-photon laser confocal scanning microscope (LSM710; Zeiss, Jena, Germany) with 630x magnification.

## Myeloperoxidase activity measurement

The myeloperoxidase (MPO) activity in intestine and peripheral blood was quantified using a commercial colorimetric assay kit (Nanjing Jiancheng Bioengineering Institute, China) according to the manufacturer's instructions. Briefly, intestinal tissues were homogenized in extraction buffer to obtain a 5% homogenate, while peripheral blood was mixed with extraction buffer at a 1:1 ratio. A 180 µl aliquot of resultant mixture was incubated with 20 µl of reaction buffer for 15 min at 37°C. Enzymatic activity was determined by measuring the changes in absorbance at 460 nm using a 96-well plate reader. MPO activity was expressed as units per gram of wet intestinal tissue or per milliliter of peripheral blood.

## Co-immunoprecipitation and western blot analysis

Co-immunoprecipitation (Co-IP) was performed to detect the interaction between Cd28/Ctla-4 and Cd80/86. HEK293T cells were co-transfected with pLVX-mCherry-C1-Cd28 (3 µg) and pEGFP-N1-Cd80/86 (3 µg) or pEGFP-N1-Ctla-4 (3 µg) and pCDNA3.1-HA-Cd80/86 (3 µg) using PEI as a transfection reagent. At 48 hr post-transfection, the cells were lysed with pre-cooled cell lysis buffer (Beyotime). The lysates were centrifuged at 12,000×g for 8 min at 4 °C, and the supernatants were incubated with mouse anti-myc mAb (Abmart) or mouse anti-HA mAb (Abmart) overnight at 4°C. Expression of the transfected plasmids was analyzed in the whole cell lysates as an input control. The following day, the mixture was incubated with 50 µl of protein A agarose beads (Thermo Fisher Scientific) for 4 hr. The beads were washed three times with lysis buffer and mixed with loading buffer for SDS-PAGE separation. Target proteins were transferred onto a 0.22 µm polyvinylidene difluoride (PVDF) membrane (EMD Millipore) for Western blot analysis. The blots were blocked with 5% skimmed milk, incubated with mouse or rabbit primary Abs overnight at 4°C, washed with TBST, and then incubated with HRP-conjugated goat anti-mouse/rabbit IgG mAb (Abmart) at room temperature for 1 hr. Detection was performed using a gel imaging system (Tanon 4500).

## Histopathological analysis

The anterior, mid, and posterior intestines (n=3 replicates) were fixed in 4% PFA overnight and embedded in paraffin. The tissues were cut into 4 µm sections and stained with hematoxylin and eosin (H&E) for histopathological analysis. To evaluate the severity of intestinal inflammation, histologic scores were determined based on established criteria from a previous study (*Erben et al., 2014*). Briefly, three independent parameters, including inflammation severity, inflammation extent, and epithelial changes, were assessed and scored as follows: inflammation severity (0=none, 1=minimal, 2=mild, 3=moderate, 4=marked); inflammation extent (0=none, 1=mucosa, 2=mucosa and submucosa, 3=transmural), epithelial changes (0=none, 1=minimal hyperplasia, 2=mild hyperplasia, minimal

goblet cell loss, 3=moderate hyperplasia, mild goblet cell loss, 4=marked hyperplasia with moderate to marked goblet cell loss). Each parameter was calculated and summed to obtain the overall score. Additionally, tissue sections were stained with Periodic Acid-Schiff (PAS) or Alcian Blue and Periodic Acid-Schiff (AB-PAS) reagent to evaluate the mucin components and goblet cell numbers.

## Transmission electron microscope observation

The posterior intestines were cut into 0.2 cm segments and then split lengthwise to expose the intestinal villi to the fixative fully. The samples were first fixed with 2.5% glutaraldehyde in phosphate buffer (0.1 M, pH 7.0) overnight at 4°C, washed three times in the phosphate buffer for 15 min at each step and post-fixed in 1% $OsO_4$ for 1 hr. Following gradient acetone dehydration and Spurr resin infiltration (1:1 and 1:3 mixture of absolute acetone and the final Spurr resin mixture for 1 hr and 3 hr, and final Spurr resin mixture overnight), the specimens were placed in an Eppendorf contained Spurr resin and heated at 70°C for overnight. The samples were sectioned using an ultratome (LEICA EM UC7). Then, the sections were stained with uranyl acetate and alkaline lead citrate for 10 min each and observed under a transmission electron microscope (Hitachi Model H-7650).

## Assessment of apoptosis by TUNEL

The posterior intestines were fixed by 4% paraformaldehyde and embedded in paraffin. Apoptosis was detected using TUNEL staining following the manufacturer's protocol (Beyotime). Briefly, deparaffinized sections were incubated with biotin-dUTP labeling solution (TdT Enzyme: Biotin-dUTP=1: 9) for 1 hr, followed by incubation with streptavidin-HRP for 30 min at room temperature. Positive signals were visualized using DAB chromogenic solution and counterstained with hematoxylin. Apoptotic cells and the area of the intestinal epithelium were quantified, and the apoptosis index was calculated as the number of apoptotic cells per $1 \times 10^4$ $\mu m^2$ observed using ImageJ software (version 1.8.0).

## RNA-sequencing and bioinformatic analysis

Total RNAs were isolated from wild-type or *ctla-4*[-/-] intestines (three biological replicates) using TRIzol reagent following the manufacturer's instructions (Takara). cDNA libraries were constructed using NEB Next Ultra Directional RNA Library Prep Kit (NEB), and sequencing was performed according to the Illumina Hiseq2500 standard protocol at LC Bio (Hangzhou, China). The differentially expressed genes (DEGs) were identified with absolute $Log_2$ fold change >1 and adjusted *p*-value < 0.05 by R package DESeq2. Gene Ontology (GO) enrichment and Kyoto encyclopedia of genes and genomes (KEGG) enrichment analyses were performed by the OmicStudio (http://www.omicstudio.cn/tool) tools. Gene-set enrichment analysis was performed using software GSEA (v4.1.0, https://www.gsea-msigdb. org/gsea/index.jsp), and the heatmap was generated using the R package ggplot2. For the protein-protein interaction (PPI) networks, the DEGs were retrieved in STRING (version 11.5, https://string-db. org/) database (combined score >0.4), and the PPI network was visualized by Cytoscape software (version 3.9.1) (*Kohl et al., 2011*). The betweenness centrality (BC) was calculated using the CytoNCA plugin in Cytoscape software. The RNA-sequencing (RNA-seq) data in this study were deposited in the Gene Expression Omnibus (GEO) (http://www.ncbi.nlm.nih.gov/geo/) database.

## Quantitative real-time PCR

The transcript abundance of target genes was quantified using quantitative real-time PCR on a CFX Connect Real-Time PCR Detection System equipped with Precision Melt Analysis Software (Bio-Rad, Cat. No. 1855200EM1). Total RNA from intestines was extracted using TRIzol reagent (Takara Bio) and reverse transcribed into cDNAs according to the manufacturer's protocol. PCR experiments were performed in a total volume of 10 µl by using an SYBR Premix Ex Taq kit (Takara Bio). The reaction mixtures were incubated for 2 min at 95°C, then subjected to 40 cycles of 15 s at 95°C, 15 s at 60°C, and 20 s at 72°C. Relative expression levels of the target genes were calculated using the $2^{-\Delta\Delta ct}$ method with β-actin for normalization. Each PCR trial was run in triplicate parallel reactions and repeated three times. The primers used in the experiments are listed in *Supplementary file 1*.

## Single-cell RNA-sequencing analysis

The intestines from wild-type (30 fish) and *ctla-4*[-/-] zebrafish (30 fish) were washed by D-Hank's and incubated with D-Hank's containing EDTA (0.37 mg/mL) and DTT (0.14 mg/mL) for 20 min. The

resulting supernatants were collected as fraction 1. The remaining tissues were subsequently digested with type IV collagenase (0.15 mg/mL) for 1 hr at room temperature and the resulting supernatants were collected as fraction 2. Fractions 1 and 2 were combined and centrifuged at 350 g for 10 min. Cells were then washed with D-Hank's and suspended in a 40% Percoll (Shanghai Yes Service Biotech, China) solution (Percoll: FBS: L-15 medium = 4: 1: 5) and passed through a 40 μm strainer (Bioland). The cell suspension was carefully layered onto 63% Percoll (Percoll: FBS: L-15 medium = 6.3: 1: 2.7) and centrifuged at 400 g for 30 min at room temperature. The cell layer at the interface was collected and washed with D-Hank's at 400 g for 10 min. Cell quantity and viability were assessed using 0.4% trypan blue staining, revealing that over 90% of the cells were viable. Single-cell samples (8,047 cells in wild-type group, 8,321 cells in *ctla-4*⁻/⁻ group) were submitted to the LC-Bio Technology Co., Ltd (Hangzhou, China) for 10x Genomics library preparation and data analysis assistance. Libraries were prepared using the Chromium Controller and Chromium Single Cell 3' Library & Gel Bead Kit v2 (10×Genomics) according to the manufacturer's protocol, and sequenced on an Illumina NovaSeq 6000 sequencing system (paired-end multiplexing run, 150 bp) at a minimum depth of 20,000 reads per cell. Sequencing results were demultiplexed and converted to FASTQ format using Illumina bcl2fastq software and the data were aligned to the Ensembl zebrafish genome assembly GRCz11. Quality control was performed using the Seurat. DoubletFinder R package was used to identify and filter the doublets (multiplets) (*McGinnis et al., 2019*). The cells were removed if they expressed fewer than 500 unique genes, or >60% mitochondrial reads. Cloud-based Cumulus v1.0 was used to perform the bath correction (using the Harmony algorithm) on the aggregated gene-count matrices (*Li et al., 2020*). The number of cells after filtration in the current study was 3263 in wild-type and 4276 in *ctla-4*⁻/⁻ groups, respectively. Cells were grouped into an optimal number of clusters for de novo cell type discovery using Seurat's `FindNeighbors()` and `FindClusters()` functions, graph-based clustering approach with visualization of cells being achieved through the use of t-SNE or UMAP plots (*Cronan et al., 2021*). The cell types were determined using a combination of marker genes identified from the literature and gene ontology. The marker genes were visualized by dot plot and t-SNE plots to reveal specific individual cell types.

## 16S rRNA gene sequencing analysis

Intestinal contents were collected from both wild-type and *ctla-4*⁻/⁻ zebrafish by gently squeezing the intestines with fine-tipped tweezers, and the remaining intestines were used for single-cell RNA-sequencing analysis. Contents from six fish were pooled to form one replicate, with each experimental sample comprising four replicates. DNA was extracted from the samples using the CTAB method, a protocol known for its efficacy in isolating DNA from trace quantities. The quality of DNA was assessed through agarose gel electrophoresis. Total DNA was amplified to construct sequencing libraries using primers (341 F: 5'-CCTACGGGNGGCWGCAG-3'; 805 R: 5'-GACTACHVGGGTATCT AATCC-3') targeting the V3-V4 regions of the 16S rRNA gene. The libraries were sequenced on an Illumina NovaSeq PE250 platform. Quality filtering was performed under specific conditions to obtain high-quality clean tags using fqtrim (v0.94). Chimeric sequences were removed using Vsearch software (v2.3.4). After dereplication using DADA2, a feature table and feature sequences were generated. Alpha diversity is applied in analyzing the complexity of species diversity for a sample through the Shannon and Simpson indices, with all calculations performed in QIIME2. Beta diversity analysis was also conducted in QIIME2, and the graphs were drawn by R package. Sequence alignment was performed using BLAST, and representative sequences were annotated using the SILVA database. Other diagrams were implemented using the R package (v3.5.2).

## In vitro lymphocyte proliferation assay

The leukocytes were prepared from the spleen, kidney, and peripheral blood of wild-type (10 fish) or *ctla-4*⁻/⁻ (10 fish) zebrafish through Ficoll-Hypaque centrifugation. A total of $2×10^6$ leukocytes from either wild-type or *ctla-4*⁻/⁻ individuals were labeled with 1 μM carboxyfluorescein succinimidyl ester (CFSE; Thermo Fisher Scientific) at 25°C for 5 min. The reaction was terminated by adding a triploid volume of Leibovitz L-15 medium (Gibco) supplemented with 10% FBS, as previously described (*Quah et al., 2007*). After washing with D-Hank's solution, the cells were cultured in L-15 medium containing 10% FBS in the presence or absence of PHA (5 μg/ml), recombinant Ctla-4-Ig (20 μg/ml), Cd28-Ig (20 μg/ml), Cd80/86 (10 μg/ml) proteins, or anti-Ctla-4 Ab (10 μg/ml) at 28°C for 3 d. CFSE

fluorescence intensity in the labeled co-cultures was analyzed using a flow cytometer (FACSJazz; BD Biosciences) to assess cell division (*Rieder et al., 2021*).

## Prediction of protein interactions by AlphaFold2

AlphaFold2 (version 2.3.2; available at https://github.com/google-deepmind/ alphafold) was implemented on a high-performance computing cluster to predict the structures of the Cd80/86 complexes with Cd28 and Ctla-4 (*Jumper et al., 2021*). The resulting models were ranked based on their per-residue Local Distance Difference Test (pLDDT) scores, which quantify the confidence level of each residue on a scale from 0 to 100. Residues were color-coded according to their pLDDT scores, with higher values reflecting greater confidence in the prediction. Furthermore, the Predicted Aligned Error (PAE) scores were analyzed to identify well-defined interaction interfaces between Cd28 or Ctla-4 and Cd80/86.

## Microscale thermophoresis assay

The binding affinity between Cd80/86 and Cd28/Ctla-4 were measured through microscale thermophoresis (MST) assays using a Monolith NT.115 instrument (Nano Temper Technologies) as previously described (*Jerabek-Willemsen et al., 2014*). In each assay, the labeled proteins (Cd28/Ctla-4 with EGFP-tag) were incubated with varying concentrations of unlabeled ligand protein (Cd80/86) for 10 min at room temperature. The initial protein concentration of 3.2 μM was diluted into 16 different concentrations by doubling dilution. The samples were then loaded into capillaries and analyzed at 25°C by using 40% light-emitting diode (LED) and medium MST power. The binding affinities of Cd80/86 with Cd28 and Cd80/86 with Ctla-4 were examined using the same parameters. Each assay was repeated three to five times, and dissociation constants ($K_D$) were calculated using MO Affinity Analysis software.

## In vivo administration of sCtla-4-Ig

An in vivo sCtla-4-Ig administration assay was conducted to evaluate the potential therapeutic effect of sCTLA-4-Ig on intervention of a *ctla-4*-deficiency induced IBD-like phenotype. For this procedure, the *ctla-4*$^{-/-}$ zebrafish were i.p administered with recombinant soluble Ctla-4-Ig protein (sCtla-4-Ig, 2 μg/g body weight) every other day until day 14. Fish that received an equal amount of human IgG isotype were used as control. The dose of sCtla-4-Ig was chosen based on its ability to significantly inhibit the mRNA expression of *il13* in Ctla-4-deficient zebrafish.

## Statistical analysis

Statistical analysis and graphical presentation were performed with GraphPad Prism 8.0. All data were presented as the mean ± SD of each group. Statistical evaluation of differences was assessed using one-way ANOVA, followed by an unpaired two-tailed *t*-test. Statistical significance was defined as *$p < 0.05$, **$p < 0.01$, ***$p < 0.001$, and ****$p < 0.0001$. All experiments were replicated at least three times.

# Acknowledgements

We are grateful to Bio-ultrastructure Analysis Laboratory at the Analysis Center of Agrobiology and Environmental Sciences, Zhejiang University, for their assistance in TEM sample preparation and observation. We also thank Hong Deng and Qiong Huang for their valuable advice and expertise in pathological analysis. Additionally, we acknowledge Shelong Zhang for his support in two-photon laser confocal scanning microscope capture. This work was supported by grants from the National Natural Science Foundation of China (32173003) and the National Key Research and Development Program of China (2018YFD0900503, 2018YFD0900505).

## Additional information

### Funding

| Funder | Grant reference number | Author |
|---|---|---|
| National Natural Science Foundation of China | 32173003 | Lixin Xiang |
| National Key Research and Development Program of China | 2018YFD0900503 | Jianzhong Shao |
| National Key Research and Development Program of China | 2018YFD0900505 | Jianzhong Shao |

The funders had no role in study design, data collection and interpretation, or the decision to submit the work for publication.

### Author contributions

Lulu Qin, Data curation, Investigation, Writing - original draft; Chongbin Hu, Software, Formal analysis, Validation; Qiong Zhao, Yong Wang, Software, Formal analysis; Dongdong Fan, Methodology; Aifu Lin, Software; Lixin Xiang, Software, Project administration, Writing – review and editing; Ye Chen, Software, Formal analysis, Project administration; Jianzhong Shao, Resources, Data curation, Formal analysis, Investigation, Visualization, Project administration, Writing – review and editing

### Author ORCIDs

Lulu Qin http://orcid.org/0009-0004-5573-2687
Chongbin Hu http://orcid.org/0000-0003-3433-4167
Ye Chen https://orcid.org/0000-0003-3671-2504
Jianzhong Shao https://orcid.org/0000-0003-3483-1817

### Ethics

All animal experiments were performed with the approval of the Ethics Committee for Animal Experimentation of Zhejiang University (Permit Number: ZJU20240828).

Reviewer #1 (Public review): https://doi.org/10.7554/eLife.101932.3.sa1
Reviewer #2 (Public review): https://doi.org/10.7554/eLife.101932.3.sa2
Reviewer #3 (Public review): https://doi.org/10.7554/eLife.101932.3.sa3
Author response https://doi.org/10.7554/eLife.101932.3.sa4

# Additional files

### Supplementary files

MDAR checklist

Supplementary file 1. The primers used in the experiments.

Supplementary file 2. The DEGs of ctla-4-/- vs WT in RNA-seq analysis.

Supplementary file 3. Gene sets for cell annotation and marker gene list of immune-cell.

Supplementary file 4. The average expression profile of cell types from zebrafish intestines.

### Data availability

RNA-seq and scRNA-seq data for this study have been deposited in NCBI Gene Expression Omnibus (GEO) (https://www.ncbi.nlm.nih.gov/geo/) under accession numbers GSE255304 and GSE255303, respectively. The 16S rRNA gene sequencing data in this study have been deposited in the NCBI Sequence Read Archive (SRA) (https://www.ncbi.nlm.nih.gov/sra/) with an accession number of BioProject PRJNA1073727.

The following datasets were generated:

| Author(s) | Year | Dataset title | Dataset URL | Database and Identifier |
|---|---|---|---|---|
| Lulu Q, Chongbin H, Ye C, Lixin X, Jianzhong S | 2024 | Ctla-4 deficiency induces an inflammatory bowel disease-like phenotype in a zebrafish model | https://www.ncbi.nlm.nih.gov/geo/query/acc.cgi?acc=GSE255303 | NCBI Gene Expression Omnibus, GSE255303 |
| Lulu Q, Chongbin H, Ye C, Lixin X, Jianzhong S | 2024 | Ctla-4 deficiency induces an inflammatory bowel disease-like phenotype in a zebrafish model | https://www.ncbi.nlm.nih.gov/geo/query/acc.cgi?acc=GSE255304 | NCBI Gene Expression Omnibus, GSE255304 |
| Lulu Q, Chongbin H, Ye C, Lixin X, Jianzhong S | 2024 | Global studies microbial diversity from zebrafish intestines | https://www.ncbi.nlm.nih.gov/bioproject/PRJNA1073727/ | NCBI BioProject, PRJNA1073727 |

The following previously published dataset was used:

| Author(s) | Year | Dataset title | Dataset URL | Database and Identifier |
|---|---|---|---|---|
| Hu C, Wang J, Hong Y, Li H, Fan D, Lin AF, Xiang LX, Shao J | 2023 | Single-cell transcriptome profiling reveals diverse immune cell populations and their responses to viral infection in the spleen of zebrafish | https://www.ncbi.nlm.nih.gov/geo/query/acc.cgi?acc=GSE211396 | NCBI Gene Expression Omnibus, GSE211396 |

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
