## [Editor Report · eLife Assessment]

This study focuses on the role of a T-cell-specific receptor, ctla-4, in a new zebrafish model of IBD-like phenotype. Although implicated in IBD diseases, the function of ctla-4 has been hard to study in mice as the KO is lethal. Ctla-4 mutant zebrafish exhibited significant intestinal inflammation and dysbiosis, mirroring the pathology of inflammatory bowel disease (IBD) in mammals, providing a new **valuable** model to the field of IBD research. This is a key study with **convincing** evidence, comprehensive transcriptomic analysis, histological examinations, and functional assays all supporting the findings.

---

## [Referee Report · Reviewer #1 (Public review)]

"Unraveling the Role of Ctla-4 in Intestinal Immune Homeostasis: Insights from a novel Zebrafish Model of Inflammatory Bowel Disease" generates a 14 bp deletion/early stop codon mutation that is viable in a zebrafish homolog of ctla-4. This mutant exhibits an IBD-like phenotype, including decreased intestinal length, abnormal intestinal folds, decreased goblet cells, abnormal cell junctions between epithelial cells, increased inflammation, and alterations in microbial diversity. Bulk and single-cell RNA-seq show upregulation of immune and inflammatory response genes in this mutant (especially in neutrophils, B cells, and macrophages) and downregulation of genes involved in adhesion and tight junctions in mutant enterocytes. The work suggests that the makeup of immune cells within the intestine is altered in these mutants, potentially due to changes in lymphocyte proliferation. Introduction of recombinant soluble Ctla-4-Ig to mutant zebrafish rescued body weight, histological phenotypes, and gene expression of several pro-inflammatory genes, suggesting a potential future therapeutic route.

Strengths:

- Generation of a useful new mutant in zebrafish ctla-4

- The demonstration of an IBD-like phenotype in this mutant is extremely comprehensive.

- Demonstrated gene expression differences provide mechanistic insight into how this mutation leads to IBD-like symptoms.

- Demonstration of rescue with a soluble protein suggests exciting future therapeutic potential

- The manuscript is mostly well organized and well written.

Initial Weaknesses were addressed during review.

---

## [Referee Report · Reviewer #2 (Public review)]

Summary:

The authors aimed to elucidate the role of Ctla-4 in maintaining intestinal immune homeostasis by using a novel Ctla-4-deficient zebrafish model. This study addresses the challenge of linking CTLA-4 to inflammatory bowel disease (IBD) due to the early lethality of CTLA-4 knockout mice. Four lines of evidence were shown to show that Ctla-4-deficient zebrafish exhibited hallmarks of IBD in mammals: (1) impaired epithelial integrity and infiltration of inflammatory cells; (2) enrichment of inflammation-related pathways and the imbalance between pro- and anti-inflammatory cytokines; (3) abnormal composition of immune cell populations; and (4) reduced diversity and altered microbiota composition. By employing various molecular and cellular analyses, the authors established ctla-4-deficient zebrafish as a convincing model of human IBD.

Strengths:

The characterization of the mutant phenotype is very thorough, from anatomical to histological and molecular levels. The finding effectively established ctla-4 mutants as a novel zebrafish model for investigating human IBD. Evidence from the histopathological and transcriptome analysis was very strong and supports a severe interruption of immune system homeostasis in the zebrafish intestine. Additional characterization using sCtla-4-Ig further probed the molecular mechanism of the inflammatory response, and provided a potential treatment plan for targeting Ctla-4 in IBD models.

Weaknesses:

To probe the molecular mechanism of Ctla-4, the authors used a spectrum of antibodies that target Ctla-4 or its receptors. The phenotype assayed was lymphocyte proliferation, while it was the composition rather than number of immune cells that was observed to be different in the scRNASeq assay. Although sCtla-4 has an effect of alleviating the IBD-like phenotypes, I found this explanation a bit oversimplified.

Comments on revised version:

The authors have sufficiently addressed all my concerns and I don't have further suggestions.

---

## [Referee Report · Reviewer #3 (Public review)]

Summary:

Current study on the mutant zebrafish for IBD modeling is worth trying. The author provided lots of evidence, including histopathological observation, gut microflora, as well as intestinal tissue or mucosa cells' transcriptomic data. The multi-omic study has demonstrated the enteritis pathology at multi levels in zebrafish model.

Strengths:

The important immune checkpoint of Treg cells were knockout in zebrafish, and the enteritis were found then. It could be a substitution of mouse knockout model to investigate the molecular mechanism of gut disease.

Weaknesses:

(1) In Fig. 2I, as to the purple glycogen signals stained by PAS was ignored for the quantitative statistics. The purple stained area could be calculated by ImageJ.

(2) Those characters in Fig. 3G are too small to recognize. It is suggested to adjusted this picture or just put it in the supplementation, with bigger size.

(3) The tissue seems damaged for IgG ctrl in Fig. 8B. It is suggested to find another slice to present here.

(4) Line 667 & 743: "16S rRNA sequencing" should be "16S rRNA gene sequencing". Please check this point throughout the text.

---

## [Author Response]

The following is the authors’ response to the original reviews

**Reviewer#1:**
The manuscript suggests the zebrafish homolog of ctla-4 and generates a new mutant in it. However, the locus that is mutated is confusingly annotated as both CD28 (current main annotation in ZFIN) and CTLA-4/CD152 (one publication from 2020), see: https://zfin.org/ZDB-GENE-070912-128. Both human CTLA-4 and CD28 align with relatively similar scores to this gene. There seem to be other orthologs of these receptors in the zebrafish genome, including CD28-like (https://zfin.org/ZDB-GENE-070912-309) which neighbors the gene annotated as CD28 (exhibiting similar synteny as human CD28 and CTLA-4). It would be helpful to provide more information to distinguish between this family of genes and to further strengthen the evidence that this mutant is in ctla-4, not cd28. Also, is one of these genes in the zebrafish genome (e.g. cd28l) potentially a second homolog of CTLA-4? Is this why this mutant is viable in zebrafish and not mammals? Some suggestions:(a) A more extensive sequence alignment that considers both CTLA-4 and CD28, potentially identifying the best homolog of each human gene, especially taking into account any regions that are known to produce the functional differences between these receptors in mammals and effectively assigns identities to the two genes annotated as "cd28" and "cd28l" as well as the gene "si:dkey-1H24.6" that your CD28 ORF primers seem to bind to in zebrafish.

In response to the reviewer's insightful suggestions, we have conducted more extensive sequence alignment and phylogenetic analyses that consider both CTLA-4, CD28, and CD28-like molecules, taking into account key regions crucial for the functionalities and functional differences between these molecules across various species, including mammals and zebrafish.

Identification of zebrafish Ctla-4: We identified zebrafish Ctla-4 as a homolog of mammalian CTLA-4 based on key conserved structural and functional characteristics. Structurally, the Ctla-4 gene shares similar exon organization compared to mammalian CTLA-4. Ctla-4 is a type I transmembrane protein with typical immunoglobulin superfamily features. Multiple amino acid sequence alignments revealed that Ctla-4 contains a ^113^LFPPPY^118^ motif and a ^123^GNGT^126^ motif in the ectodomain, and a tyrosine-based ^206^YVKF^209^ motif in the distal C-terminal region. These motifs closely resemble MYPPPY, GNGT, and YVKM motifs in mammalian CTLA-4s, which are essential for binding to CD80/CD86 ligands and molecular internalization and signaling inhibition. Despite only 23.7% sequence identity to human CTLA-4, zebrafish Ctla-4 exhibits a similar tertiary structure with a two-layer β-sandwich architecture in its extracellular IgV-like domain. Four cysteine residues responsible for the formation of two pairs of disulfide bonds (Cys^20^-Cys^91^/Cys^46^-Cys^65^ in zebrafish and Cys^21^-Cys^92^/Cys^48^-Cys^66^ in humans) that connect the two-layer β-sandwich are conserved. Additionally, a separate cysteine residue (Cys^120^ in zebrafish and Cys^120^ in humans) involved in dimerization is also present, and Western blot analysis under reducing and non-reducing conditions confirmed Ctla-4’s dimerization. Phylogenetically, Ctla-4 clusters with other known CTLA-4 homologs from different species with high bootstrap probability, while zebrafish Cd28 groups separately with other CD28s. Functionally, Ctla-4 is predominantly expressed on CD4^+^ T and CD8^+^ T cells in zebrafish. It plays a pivotal inhibitory role in T cell activation by competing with CD28 for binding to CD80/86, as validated through a series of both in vitro and in vivo assays, including microscale thermophoresis assays which demonstrated that Ctla-4 exhibits a significantly higher affinity for Cd80/86 than Cd28 (KD = 0.50 ± 0.25 μM vs. KD = 2.64 ± 0.45 μM). These findings confirm Ctla-4 as an immune checkpoint molecule, reinforcing its identification within the CTLA-4 family.

Comparison between zebrafish Cd28 and "Cd28l": Zebrafish Cd28 contains an extracellular SYPPPF motif and an intracellular FYIQ motif. The extracellular SYPPPF motif is essential for binding to Cd80/CD86, while the intracellular FYIQ motif likely mediates kinase recruitment and co-stimulatory signaling. In contrast, the "Cd28l" molecule lacks the SYPPPF motif, which is critical for Cd80/CD86 binding, and exhibits strong similarity in its C-terminal 79 amino acids to Ctla-4 rather than Cd28. Consequently, "Cd28l" resembles an atypical Ctla-4-like molecule but fails to exhibit Cd80/CD86 binding activity.

We have incorporated the relevant analysis results into the main text of the revised manuscript and updated Supplementary Figure 1. Additionally, we provide key supplementary analyses here for the reviewer's convenience.

**Author response image 1. sa4fig1:** Illustrates the alignment of Ctla-4 (XP_005167576.1) and Ctla-4-like (XP_005167567.1, previously referred to as ‘Cd28l’) in zebrafish, generated using ClustalX and Jalview. Conserved and partially conserved amino acid residues are highlighted in color gradients ranging from carnation to red, respectively. The B7-binding motif is encircled with a red square.

(b) Clearer description in the main text of such an analysis to better establish that the mutated gene is a homolog of ctla-4, NOT cd28.

We appreciate the reviewer's advice. Additional confirmation of zebrafish Ctla-4 is detailed in lines 119-126 of the revised manuscript.

(c) Are there mammalian anti-ctla-4 and/or anti-cd28 antibodies that are expected to bind to these zebrafish proteins? If so, looking to see whether staining is lost (or western blotting is lost) in your mutants could be additionally informative. (Our understanding is that your mouse anti-Ctla-4 antibody is raised against recombinant protein generated from this same locus, and so is an elegant demonstration that your mutant eliminates the production of the protein, but unfortunately does not contribute additional information to help establish its homology to mammalian proteins).

This suggestion holds significant value. However, a major challenge in fish immunology research is the limited availability of antibodies suitable for use in fish species; antibodies developed for mammals are generally not applicable. We attempted to use human and mouse anti-CTLA-4 and anti-CD28 antibodies to identify Ctla-4 and Cd28 in zebrafish, but the results were inconclusive, with no expected signals. This outcome likely arises from the low sequence identity between human/mouse CTLA-4 and CD28 and their zebrafish homologs (ranging from 21.3% to 23.7% for CTLA-4 and 21.2% to 24.0% for CD28). Therefore, developing specific antibodies against zebrafish Ctla-4 is essential for advancing this research.

The methods section is generally insufficient and doesn't describe many of the experiments performed in this manuscript. Some examples:(a) No description of antibodies used for staining or Western blots (Figure1C, 1D, 1F).(b) No description of immunofluorescence protocol (Figure 1D, 1F).(c) No description of Western blot protocol (Figure 1C, 2C).(d) No description of electron microscopy approach (Figure 2K).(e) No description of the approach for determining microbial diversity (Entirety of Figure 6).(f) No description of PHA/CFSE/Flow experiments (Figure 7A-E).(g) No description of AlphaFold approach (Figures 7F-G).(h) No description of co-IP approach (Figure 7H).(i) No description of MST assay or experiment (Figure 7I).(j) No description of purification of recombinant proteins, generation of anti-Ctla-4 antibody, or molecular interaction assays (Figures S2 and S6).

We apologize for this oversight. The methods section was inadvertently incomplete due to an error during the file upload process at submission. This issue has been addressed in the revised manuscript. We appreciate your understanding.

Figure 5 suggests that there are more Th2 cells 1, Th2 cells 2, and NKT cells in ctla-4 mutants through scRNA-seq. However, as the cell numbers for these are low in both genotypes, there is only a single replicate for each genotype scRNA-seq experiment, and dissociation stress can skew cell-type proportions, this finding would be much more convincing if another method that does not depend on dissociation was used to verify these results. Furthermore, while Th2 cells 2 are almost absent in WT scRNA-seq, KEGG analysis suggests that a major contributor to their clustering may be ribosomal genes (Fig. 5I). Since no batch correction was described in the methods, it would be beneficial to verify the presence of this cluster in ctla-4 mutants and WT animals through other means, such as in situ hybridization or transgenic lines.

We are grateful for the insightful comments provided by the reviewer. Given that research on T cell subpopulations in fish is still in its nascent stages, the availability of specific marker antibodies and relevant transgenic strains remains limited. Our single-cell RNA sequencing (scRNA-seq) analysis revealed that a distinct Th2 subset 2 was predominantly observed in Ctla-4 mutants but was rare in wild-type zebrafish, it suggests that this subset may primarily arise under pathological conditions associated with Ctla-4 mutation. Due to the near absence of Th2 subset 2 in wild-type samples, KEGG enrichment analysis was performed exclusively on this subset from Ctla-4-deficient intestines. The ribosome pathway was significantly enriched, suggesting that these cells may be activated to fulfill their effector functions. However, confirming the presence of Th2 subset 2 using in situ hybridization or transgenic zebrafish lines is currently challenging due to the lack of lineage-specific markers for detailed classification of Th2 cell subsets and the preliminary nature of scRNA-seq predictions.

To address the reviewers' suggestion to confirm compositional changes in Th2 and NKT cells using dissociation-independent methods, we quantified mRNA levels of Th2 (il4, il13, and gata3) and NKT (nkl.2, nkl.4, and prf1.1) cell marker genes via RT-qPCR in intestines from wild-type and mutant zebrafish. As shown in Figure S7B and S7C, these markers were significantly upregulated in Ctla-4-deficient intestines compared to wild-type controls. This indicates an overall increase in Th2 and NKT cell activity in mutant zebrafish, aligning with our scRNA-seq analysis and supports the validity of our initial findings.

Before analyzing the scRNA-seq data, we performed batch correction using the Harmony algorithm via cloud-based Cumulus v1.0 on the aggregated gene-count matrices. This methodological detail has been included in the “Materials and Methods” section of the revised manuscript. Moreover, the RT-qPCR results are presented in Supplementary Figures S7B and S7C.

Quality control (e.g., no. of UMIs, no. of genes, etc.) metrics of the scRNAseq experiments should be presented in the supplementary information for each sample to help support that observed differential expression is not merely an outcome of different sequencing depths of the two samples.

As illustrated in Fig. S5, the quality control data have been supplemented to include the effective cell number of the sample, along with pre- and post-filtering metrics such as nFeature_RNA, nCount_RNA and mitochondrial percentage (percent.mito). Furthermore, scatter plots comparing the basic information of the sample cells before and after filtering are provided.

Some references to prior research lack citations. Examples:(a)"Given that Ctla-4 is primarily expressed on T cells (Figure 1E-F), and its absence has been shown to result in intestinal immune dysregulation, indicating a crucial role of this molecule as a conserved immune checkpoint in T cell inhibition."

The references were incorporated into line 71 of the revised manuscript.

(b) Line 83: Cite evidence/review for the high degree of conservation in adaptive immunity.

The references were incorporated into line 93 of the revised manuscript.

(c) Lines 100-102: Cite the evidence that MYPPPY is a CD80/86 binding motif.

The references were incorporated into line 117 of the revised manuscript.

The text associated with Figure 8 (Lines 280-289) does not clearly state that rescue experiments are being done in mutant zebrafish.

We have provided a clear explanation of the rescue experiments conducted in Ctla-4-deficient zebrafish. This revision has been incorporated into line 319.

Line 102: Is there evidence from other animals that LFPPPY can function as a binding site for CD80/CD86? Does CD28 also have this same motif?

The extracellular domains of CTLA-4 and CD28, which bind to CD80/CD86, are largely conserved across various species. This conservation is exemplified by a central PPP core motif, although the flanking amino acids exhibit slight variations. In mammals, both CTLA-4 and CD28 feature the conserved MYPPPY motif. By contrast, in teleost fish, such as rainbow trout, CTLA-4 contains an LYPPPY motif, while CD28 has an MYPPPI motif (Ref. 1). Grass carp CTLA-4 displays an LFPPPY motif, whereas its CD28 variant bears an IYPPPF motif. Yeast two-hybrid assays confirm that these motifs facilitate interactions between grass carp CTLA-4 and CD28 with CD80/CD86 (Ref. 2). Similarly, zebrafish Ctla-4 contains the LFPPPY motif observed in grass carp, while Cd28 exhibits a closely related SYPPPF motif.

References:

(1) Bernard, D et al. (2006) Costimulatory Receptors in a Teleost Fish: Typical CD28, Elusive CTLA-4. J Immunol. 176: 4191-4200.

(2) Lu T Z et al. (2022) Molecular and Functional Analyses of the Primordial Costimulatory Molecule CD80/86 and Its Receptors CD28 and CD152 (CTLA-4) in a Teleost Fish. Frontiers in Immunology. 13:885005.

Line 110-111: Suggest adding citation of these previously published scRNAseq data to the main text in addition to the current description in the Figure legend.

The reference has been added in line 129 in the main text.

Figure 3B: It would be helpful to label a few of the top differentially expressed genes in Panel B?

The top differentially expressed genes have been labeled in Figure 3B.

Figure 3G: It's unclear how this analysis was conducted, what this figure is supposed to demonstrate, and in its current form it is illegible.

Figure 3G displays a protein-protein interaction network constructed from differentially expressed genes. The densely connected nodes, representing physical interactions among proteins, provide valuable insights for basic scientific inquiry and biological or biomedical applications. As proteins are crucial to diverse biological functions, their interactions illuminate the molecular and cellular mechanisms that govern both healthy and diseased states in organisms. Consequently, these networks facilitate the understanding of pathogenic and physiological processes involved in disease onset and progression.

To construct this network, we first utilized the STRING database (https://string-db.org) to generate an initial network diagram using the differentially expressed genes. This diagram was subsequently imported into Cytoscape (version 3.9.1) for visualization and further analysis. Node size and color intensity reflect the density of interactions, indicating the relative importance of each protein. Figure 3G illustrates that IL1β was a central cytokine hub in the disease process of intestinal inflammation in Ctla-4-deficient zebrafish.

Expression scale labeling:(a) Most gene expression scales are not clearly labeled: do they represent mean expression or scaled expression? Has the expression been log-transformed, and if so, which log (natural log? Log10? Log2?). See: Figure 3E, 3I, 4D, 4E, 5B, 5G, 5H, 6I.

The gene expression scales are detailed in the figure legends. Specifically, Figures 3E, 3I, and 6I present heatmaps depicting row-scaled expression levels for the corresponding genes. In contrast, Figures 4D and 4E display heatmaps illustrating the mean expression of these genes. Additionally, the dot plots in Figures 5B, 5G, and 5H visualize the mean expression levels of the respective genes.

(b) For some plots, diverging color schemes (i.e. with white/yellow in the middle) are used for non-diverging scales and would be better represented with a sequential color scale. See: 4D, 4E, and potentially others (not fully clear because of the previous point).

The color schemes in Figures 4D and 4E have been updated to a sequential color scale. The gene expression data depicted in these figures represent mean expression values and have not undergone log transformation. This information has been incorporated into the figure legend for clarity.

Lines 186-187: Though it is merely suggested, apoptotic gene expression can be upregulated as part of the dissociation process for single-cell RNAseq. This would be much stronger if supported by a staining, such as anti-Caspase 3.

Following the reviewer's insightful recommendations, we conducted a TUNEL assay to evaluate apoptosis in the posterior intestinal epithelial cells of both wild-type and Ctla-4-deficient zebrafish. As expected, our results demonstrate a significant increase in epithelial cell apoptosis in Ctla-4-deficient zebrafish compared with wild-type fish. The corresponding data are presented in Figure S6D and have been incorporated into the manuscript. Detailed protocols for the TUNEL assay have also been included in the Materials and Methods section.

**Author response image 2. sa4fig2:** Illustrates the quantification of TUNEL-positive cells per 1 × 104 μm2/⁻ in the posterior intestines of both wild-type (WT) and ctla-4⁻/⁻ zebrafish (n = 5). The data demonstrate a comparative analysis of apoptotic cell density between the two genotypes.

Lines 248-251: This manuscript demonstrates gut inflammation and also changes in microbial diversity, but I don't think it demonstrates an association between them, which would require an experiment that for instance rescues one of these changes and shows that it ameliorates the other change, despite still being a ctla-4 mutant.

We appreciate the valuable comments from the reviewer. Recently, the relationship between inflammatory bowel disease (IBD) and gut microbial diversity has garnered considerable attention, with several key findings emerging from human IBD studies. For instance, patients with IBD (including ulcerative colitis and Crohn's disease) exhibit reduced microbial diversity, which is correlated with disease severity. This decrease in microbial richness is thought to stem from the loss of normal anaerobic bacteria, such as *Bacteroides*, *Eubacterium*, and *Lactobacillus* (Refs. 1-6). Research using mouse models has shown that inflammation increases oxygen and nitrate levels within the intestinal lumen, along with elevated host-derived electron acceptors, thereby promoting anaerobic respiration and overgrowth of *Enterobacteriaceae* (Ref 7). Consistent with these findings, our study observed a significant enrichment of *Enterobacteriaceae* in the inflamed intestines of Ctla-4-deficient zebrafish, which supporting the observations in mice. Despite this progress, the zebrafish model for intestinal inflammation remains under development, with limitations in available techniques for manipulating intestinal inflammation and reconstructing gut microbiota. These challenges hinder investigations into the association between intestinal inflammation and changes in microbial diversity. We plan to address these issues through ongoing technological advancements and further research. We thank the reviewer for their understanding.

References:

(1) Ott S J, Musfeldt M, Wenderoth D F, Hampe J, Brant O, Fölsch U R *et al*. (2004) Reduction in diversity of the colonic mucosa associated bacterial microflora in patients with active inflammatory bowel disease. Gut 53:685-693.

(2) Manichanh C, Rigottier-Gois L, Bonnaud E, Gloux K, Pelletier E, Frangeul L et al. (2006) Reduced diversity of faecal microbiota in Crohn's disease revealed by a metagenomic approach. Gut 55:205-211.

(3) Qin J J, Li R Q, Raes J, Arumugam M, Burgdorf K S, Manichanh C et al. (2010) A human gut microbial gene catalogue established by metagenomic sequencing. Nature 464:59-U70.

(4) Sha S M, Xu B, Wang X, Zhang Y G, Wang H H, Kong X Y et al. (2013) The biodiversity and composition of the dominant fecal microbiota in patients with inflammatory bowel disease. Diagn Micr Infec Dis 75:245-251.

(5) Ray K. (2015) IBD. Gut microbiota in IBD goes viral. Nat Rev Gastroenterol Hepatol 12:122.

(6) Papa E, Docktor M, Smillie C, Weber S, Preheim S P, Gevers D et al. (2012) Non-Invasive Mapping of the Gastrointestinal Microbiota Identifies Children with Inflammatory Bowel Disease. Plos One 7: e39242-39254.

(7) Hughes E R, Winter M G, Duerkop B A, Spiga L, de Carvalho T F, Zhu W H et al. (2017) Microbial Respiration and Formate Oxidation as Metabolic Signatures of Inflammation-Associated Dysbiosis. Cell Host Microbe 21:208-219.

Lines 270-272 say that interaction between Cd28/ctla-4 and Cd80/86 was demonstrated through bioinformatics, flow-cytometry, and Co-IP. Does this need to reference Fig S6D for the flow data? Figures 7F-G are very hard to read or comprehend as they are very small. Figure 7H is the most compelling evidence of this interaction and might stand out better if emphasized with a sentence referencing it on its own in the manuscript.

In this study, we utilized an integrated approach combining bioinformatics prediction, flow cytometry, and co-immunoprecipitation (Co-IP) to comprehensively investigate and validate the interactions between Cd28/Ctla-4 and Cd80/86. Flow cytometry analysis, as depicted in Supplementary Figure 6D (revised as Supplementary Figure 8F), demonstrated the surface expression of Cd80/86 on HEK293T cells and quantified their interactions with Cd28 and Ctla-4. These experiments not only validated the interactions between Cd80/86 and Cd28/Ctla-4 but also revealed a dose-dependent relationship, providing robust supplementary evidence for the molecular interactions under investigation. Furthermore, in Figure 7F-G, the axis font sizes were enlarged to improve readability. Additionally, in response to reviewers' feedback, we have emphasized Figure 7H, which presents the most compelling evidence for molecular interactions, by including a standalone sentence in the text to enhance its prominence.

For Figure 7A-E, for non-immunologists, it is unclear what experiment was performed here - it would be helpful to add a 1-sentence summary of the assay to the main text or figure legend.

We apologize for this oversight. Figures 7A–E illustrate the functional assessment of the inhibitory role of Ctla-4 in Cd80/86 and Cd28-mediated T cell activation. A detailed description of the methodologies associated with Figures 7A–E is provided in the ‘Materials and Methods’ section of the revised manuscript.

For Figure 7F-G, it is extremely hard to read the heat map legends and the X and Y-axis. Also, what the heatmaps show and how that fits the overall narrative can be elaborated significantly.

We regret this oversight. To enhance clarity, we have increased the font size of the heatmap legends and the X and Y-axes, as shown in the following figure. Additionally, a detailed analysis of these figures is provided in lines 299–306 of the main text.

In general, the main text that accompanies Figure 7 should be expanded to more clearly describe these experiments/analyses and their results.

We have conducted a detailed analysis of the experiments and results presented in Figure 7. This analysis is described in lines 278-314.

**Reviewer #2:**
The scRNASeq assay is missing some basic characterization: how many WT and mutant fish were assayed in the experiment? how many WT and mutant cells were subject to sequencing? Before going to the immune cell types, are intestinal cell types comparable between the two conditions? Are there specific regions in the tSNE plot in Figure 4A abundant of WT or ctla-4 mutant cells?

In the experiment, we analyzed 30 wild-type and 30 mutant zebrafish for scRNA-seq, with an initial dataset comprising 8,047 cells in the wild-type group and 8,321 cells in the mutant group. Sample preparation details are provided on lines 620-652. Due to the relatively high expression of mitochondrial genes in intestinal tissue, quality control filtering yielded 3,263 cells in the wild-type group and 4,276 cells in the mutant group. Given that the intestinal tissues were dissociated using identical protocols, the resulting cell types are comparable between the two conditions. Both the wild-type and Ctla-4-deficient groups contained enterocytes, enteroendocrine cells, smooth muscle cells, neutrophils, macrophages, B cells, and a cluster of T/NK/ILC-like cells. Notably, no distinct regions were enriched for either condition in the tSNE plot (Figure 4A).

The cell proliferation experiment using PHA stimulation assay demonstrated the role of Ctla-4 in cell proliferation, while the transcriptomic evidence points towards activation rather than an overall expansion of T-cell numbers. This should be discussed towards a more comprehensive model of how subtypes of cells can be differentially proliferating in the disease model.

In the PHA-stimulated T cell proliferation assay, we aimed to investigate the regulatory roles of Ctla-4, Cd28, and Cd80/86 in T cell activation, focusing on validating Ctla-4's inhibitory function as an immune checkpoint. While our study examined general regulatory mechanisms, it did not specifically address the distinct roles of Ctla-4 in different T cell subsets. We appreciate the reviewer's suggestion to develop a more comprehensive model that elucidates differential T cell activation across various subsets in disease models. However, due to the nascent stage of research on fish T cell subsets and limitations in lineage-specific antibodies and transgenic strains, such investigations are currently challenging. We plan to pursue these studies in the future. Despite these constraints, our single-cell RNA sequencing data revealed an increased proportion of Th2 subset cells in Ctla-4-deficient zebrafish, as evidenced by elevated expression levels of Th2 markers (Il4, Il13, and Gata3) via RT-qPCR (see Figures S7B). Notably, recent studies in mouse models have shown that naïve T cells from CTLA-4-deficient mice tend to differentiate into Th2 cells post-proliferation, with activated Th2 cells secreting higher levels of cytokines like IL-4, IL-5, and IL-13, thereby exerting their effector functions (Refs. 1-2). Consequently, our findings align with observations in mice, suggesting conserved CTLA-4 functions across species. We have expanded the "Discussion" section to clarify these points.

References:

(1) Bour-Jordan H, Grogan J L, Tang Q Z, Auger J A, Locksley R M, Bluestone J A et al. (2003) CTLA-4 regulates the requirement for cytokine-induced signals in T_H_2 lineage commitment. Nature Immunology 4: 182-188.

(2) Khattri Roli, Auger, Julie A, Griffin Matthew D, Sharpe Arlene H, Bluestone Jeffrey A et al. (1999) Lymphoproliferative Disorder in CTLA-4 Knockout Mice Is Characterized by CD28-Regulated Activation of Th2 Responses. The Journal of Immunology 162:5784-5791.

It would be nice if the authors could also demonstrate whether other tissues in the zebrafish have an inflammation response, to show whether the model is specific to IBD.

In addition to intestinal tissues, we also performed histological analysis on the liver of Ctla-4-deficient zebrafish. The results showed that Ctla-4 deficiency led to mild edema in a few hepatocytes, and lymphocyte infiltration was not significant. Compared to the liver, we consider intestinal inflammation to be more pronounced.

Some minor comments on terminology(a) "multiomics" usually refers to omics experiments with different modalities (e.g. transcriptomics, proteomics, metabolomics etc), while the current paper only has transcriptomics assays. I wouldn't call it "multiomics" analysis.

We appreciate the reviewer's attention to this issue. The "multi-omics" has been revised to "transcriptomics".

(b) In several parts of the figure legend the author mentioned "tSNE nonlinear clustering" (Figures 4A and 5A). tSNE is an embedding method rather than a clustering method.

The "tSNE nonlinear clustering" has been revised to "tSNE embedding”.

(c) Figure 1E is a UMAP rather than tSNE.

The "tSNE" has been revised to "UMAP" in the figure legend in line 1043.

**Reviewer#3:**
Line 28: The link is not directly reflected in this sentence describing CTLA-4 knockout mice.

We appreciate the reviewer for bringing this issue to our attention. We have expanded our description of CTLA-4 knockout mice on lines 77-84.

Line 80-83: There is a lack of details about the CTLA-4-deficient mice. The factor that Th2 response could be induced has been revealed in mouse model. See the reference entitled "CTLA-4 regulates the requirement for cytokine-induced signals in TH2 lineage commitment" published in Nature Immunology.

We thank the reviewer for providing valuable references. We have added descriptions detailing the differentiation of T cells into Th2 cells in CTLA-4-deficient mice on lines 78–81, and the relevant references have been cited in the revised manuscript.

To better introduce the CTLA-4 immunobiology, the paper entitled "Current Understanding of Cytotoxic T Lymphocyte Antigen-4 (CTLA-4) Signaling in T-Cell Biology and Disease Therapy" published in Molecules and Cells should be referred.

We have provided additional details on CTLA-4 immunology (lines 75-84) and have included the relevant reference in the revised manuscript.

In current results, there are many sentences that should be moved to the discussion, such as lines 123-124, lines 152-153, lines 199-200, and lines 206-207. So, the result sections just describe the results, and the discussions should be put together in the discussion.

We have relocated these sentences to the 'Discussion' section and refined the writing.

In the discussion, the zebrafish enteritis model, such as DSS/TNBS and SBMIE models, should also be compared with the current CTLA-4 knockout model. Also, the comparison between the current fish IBD model and the previous mouse model should also be included, to enlighten the usage of CTLA-4 knockout zebrafish IBD model.

We compared the phenotypes of our current Ctla-4-knockout zebrafish IBD model with other models, including DSS-induced IBD models in zebrafish and mice, as well as TNBS- and SBM-induced IBD models in zebrafish. The details are included in the "Discussion" section (lines 353-365).

As to the writing, the structure of the discussion is poor. The paragraphs are very long and hard to follow. Many findings from current results were not yet discussed. I just can't find any discussion about the alteration of intestinal microbiota.

In response to the reviewers' constructive feedback, we have revised and enhanced the discussion section. Furthermore, we have integrated the most recent research findings relevant to this study into the discussion to improve its relevance and comprehensiveness.

In the discussion, the aerobic-related bacteria in 16s rRNA sequencing results should be focused on echoing the histopathological findings, such as the emptier gut of CTLA-4 knockout zebrafish.

As mentioned above, the discussion section has been revised and expanded to provide a better understanding of the potential interplay among intestinal inflammatory pathology, gut microbiota alterations, and immune cell dysregulation in Ctla-4-deficient zebrafish. Furthermore, promising avenues for future research that warrant further investigation were also discussed.

In the current method, there are no descriptions for many used methods, which already generated results, such as WB, MLR, MST, Co-IP, AlphaFold2 prediction, and how to make currently used anti-zfCTLA4 antibody. Also, there is a lack of description of the method of the husbandry of knockout zebrafish line.

We regret these flaws. The methods section was inadvertently incomplete due to an error during the file upload process at submission. This issue has been rectified in the revised manuscript. Additionally, Ctla-4-deficient zebrafish were reared under the same conditions as wild-type zebrafish, and the rearing methods are now described in the "Generation of Ctla-4-deficient zebrafish" section of the Materials and Methods.

Line 360: the experimental zebrafish with different ages could be a risk for unstable intestinal health. See the reference entitled "The immunoregulatory role of fish-specific type II SOCS via inhibiting metaflammation in the gut-liver axis" published in Water Biology and Security. The age-related differences in zebrafish could be observed in the gut.

We appreciate the reviewers' reminders. The Ctla-4 mutant zebrafish used in our experiments were 4 months old, while the wild-type zebrafish ranged from 4 to 6 months old. These experimental fish were relatively young and uniformly distributed in age. During our study, we examined the morphological structures of the intestines in zebrafish aged 4 to 6 months and observed no significant abnormalities. These findings align with previous research indicating no significant difference in intestinal health between 3-month-old and 6-month-old wild-type zebrafish (Ref. 1). Consequently, we conclude that there is no notable aging-related change in the intestines of zebrafish aged 4 to 6 months. This reduces the risk associated with age-related variables in our study. We have added an explanation stating that the Ctla-4 mutant zebrafish used in the experiments were 4 months old (Line 449) in the revised manuscript.

Reference

(1) Shan Junwei, Wang Guangxin, Li Heng, Zhao Xuyang et al. (2023) The immunoregulatory role of fish-specific type II SOCS via inhibiting metaflammation in the gut-liver axis. Water Biology and Security 2: 100131-100144.

Section "Generation of Ctla-4-deficient zebrafish": There is a lack of description of PCR condition for the genotyping.

The target DNA sequence was amplified at 94 °C for 4 min, followed by 35 cycles at 94°C for 30 s, 58°C for 30 s and 72°C for 30 s, culminating in a final extension at 72 °C for 10 min. The polymerase chain reaction (PCR) conditions are described in lines 458-460.

How old of the used mutant fish? There should be a section "sampling" to provide the sampling details.

The "Sampling" information has been incorporated into the "Materials and Methods" section of the revised manuscript. Wild-type and Ctla-4-deficient zebrafish of varying months were housed in separate tanks, each labeled with its corresponding birth date. Experiments utilized Ctla-4-deficient zebrafish aged 4 months and wild-type zebrafish aged between 4 to 6 months.

Line 378-380: The index for the histopathological analysis should be detailed, rather than just provide a reference. I don't think these indexes are good enough to specifically describe the pathological changes of intestinal villi and mucosa. It is suggested to improve with detailed parameters. As described in the paper entitled "Pathology of Gastric Intestinal Metaplasia: Clinical Implications" published in Am J Gastroenterol., histochemical, normal gastric mucins are pH neutral, and they stain magenta with periodic acid-Schiff (PAS). In an inflamed gut, acid mucins replace the original gastric mucins and are stained blue with Alcian blue (AB). So, to reveal the pathological changes of goblet cells and involved mucin components, AB staining should be added. Also, for the number of goblet cells in the inflammatory intestine, combining PAS and AB staining is the best way to reveal all the goblet cells. In Figure 2, there were very few goblet cells. The infiltration of lymphocytes and the empty intestinal lumen could be observed. Thus, the ratio between the length of intestinal villi and the intestinal ring radius should calculated.

In response to the reviewers’ valuable suggestions, we have augmented the manuscript by providing additional parameters related to the pathological changes observed in the Ctlta-4-deficient zebrafish intestines, including the mucin component changes identified through PAS and AB-PAS staining, the variations in the number of goblet cells evaluated by AB-PAS staining, and the ratio of intestinal villi length to the intestinal ring radius, as illustrated in the following figures. These new findings are detailed in the "Materials and Methods" (lines 563-566) and "Results" (lines 143-146) sections, along with Supplementary Figure S3 of the revised manuscript.

Section "Quantitative real-time PCR": What's the machine used for qPCR? How about the qPCR validation of RNA seq data? I did not see any related description of data and methods for qPCR validation. In addition, beta-actin is not a stable internal reference gene, to analyze inflammation and immune-related gene expression. See the reference entitled "Actin, a reliable marker of internal control?" published in Clin Chim Acta. Other stable housekeeping genes, such as EF1alpha and 18s, could be better internal references.

RT-qPCR experiments were conducted using a PCR thermocycler device (CFX Connect Real-Time PCR Detection System with Precision Melt Analysis Software, Bio-Rad, Cat. No. 1855200EM1). This information has been incorporated into lines 608-610 of the "Materials and Methods" section. In these experiments, key gene sequences of interest, including il13, mpx, and il1β, were extracted from RNA-seq data for RT-qPCR validation. To ensure accurate normalization, potential internal controls were evaluated, and β-actin was identified as a suitable candidate due to its consistent expression levels in the intestines of both wild-type and Ctla-4-deficient zebrafish. The use of β-actin as an internal control is further supported by its application in recent studies on intestinal inflammation (Refs 1–2).

References:

(1) Tang Duozhuang, Zeng Ting, Wang Yiting, Cui Hui et al. (2020) Dietary restriction increases protective gut bacteria to rescue lethal methotrexate-induced intestinal toxicity. Gut Microbes 12: 1714401-1714422.

(2) Malik Ankit, Sharma Deepika et al. (2023) Epithelial IFNγ signaling and compartmentalized antigen presentation orchestrate gut immunity. Nature 623: 1044-1052.

How to generate sCtla-4-Ig, Cd28-Ig and Cd80/86? No method could be found.

We apologize for the omission of these methods. The detailed protocols have now been added to the "Materials and Methods" section of the revised manuscript (lines 464-481).

Figure 5: As reviewed in the paper entitled "Teleost T and NK cell immunity" published in Fish and Shellfsh Immunology, two types of NK cell homologues have been described in fish: non-specific cytotoxic cells and NK-like cells. There is no NKT cell identified in the teleost yet. Therefore, "NKT-like" could be better to describe this cell type.

We refer to "NKT" cells as "NKT-like" cells, as suggested.

For the supplementary data of scRNA-seq, there lacks the details of expression level.

The expression levels of the corresponding genes are provided in Supplemental Table 4.

Supplemental Table 1: There are no accession numbers of amplified genes.

The accession numbers of the amplified genes are included in Supplemental Table 1.

The English needs further editing.

We have made efforts to enhance the English to meet the reviewers' expectations.

Line 32: The tense should be the past.

This tense error has been corrected.

Line 363-365: The letter of this approval should be provided as an attachment.

The approval document is provided as an attachment.

Line 376: How to distinguish the different intestinal parts? Were they judged as the first third, second third, and last third parts of the whole intestine?

The differences among the three segments of zebrafish intestine are apparent. The intestinal tube narrows progressively from the anterior to the mid-intestine and then to the posterior intestine. Moreover, the boundaries between the intestinal segments are well-defined, facilitating the isolation of each segment.

Line 404: Which version of Cytoscape was used?

The version of Cytoscape used in this study is 3.9.1. Information about the Cytoscape version is provided on line 603.

The product information of both percoll and cell strainer should be provided.

The information regarding Percoll and cell strainers has been added on lines 626 and 628, respectively.

Line 814: Here should be a full name to tell what is MST.

The acronym MST stands for "Microscale Thermophoresis", a technique that has been referenced on lines 1157-1158.